# Learning Spatial-Aware Manipulation Ordering

**Yuxiang Yan**[1,*]    **Zhiyuan Zhou**[1,*]    **Xin Gao**[1]    **Guanghao Li**[1]
**Shenglin Li**[2]    **Jiaqi Chen**[3]    **Qunyan Pu**[2]    **Jian Pu**[1,†]

[1] Fudan University    [2] Shanghai YinCheng Intelligent CO., LTD    [3] Stanford University

{yxyan22, zhouzy25}@m.fudan.edu.cn,   jianpu@fudan.edu.cn

## Abstract

Manipulation in cluttered environments is challenging due to spatial dependencies among objects, where an improper manipulation order can cause collisions or blocked access. Existing approaches often overlook these spatial relationships, limiting their flexibility and scalability. To address these limitations, we propose OrderMind, a unified spatial-aware manipulation ordering framework that directly learns object manipulation priorities based on spatial context. Our architecture integrates a spatial context encoder with a temporal priority structuring module. We construct a spatial graph using k-Nearest Neighbors to aggregate geometric information from the local layout and encode both object-object and object-manipulator interactions to support accurate manipulation ordering in real-time. To generate physically and semantically plausible supervision signals, we introduce a spatial prior labeling method that guides a vision-language model to produce reasonable manipulation orders for distillation. We evaluate OrderMind on our Manipulation Ordering Benchmark, comprising 163,222 samples of varying difficulty. Extensive experiments in both simulation and real-world environments demonstrate that our method significantly outperforms prior approaches in effectiveness and efficiency, enabling robust manipulation in cluttered scenes.

## 1 Introduction

Object manipulation [1–3] in real-world environments is a fundamental capability for robots to interact with diverse objects to perform tasks [4]. In cluttered environments, objects are often closely packed, partially occluded, or physically constrained by one another [5]. The order in which objects are manipulated can significantly impact both task efficiency and scene stability. Improper ordering may lead to collisions or structural collapse [6]. Effective manipulation ordering requires understanding how the spatial layout of objects constrains their interactions. Capturing spatial conditions and inferring manipulation order is essential for enabling safe robotic manipulation.

Most existing approaches for manipulation in cluttered environments do not explicitly model manipulation ordering. As shown in Fig. 1(a), some systems [7–9] systems apply heuristic algorithm after object recognition. These pipelines are generally limited in their ability to generalize and tend to break down when spatial relations vary across different scenes. More recent methods, shown in Fig. 1(b), leverage large vision-language models to guide the manipulation sequence through explicit prompts. While these approaches are flexible, they suffer from long inference times [10], often spanning several seconds, making them unsuitable for real-time deployment. These limitations motivate the need for a unified framework that can directly learn spatial-aware manipulation ordering with low latency and high adaptability.

---

*These authors contributed equally to this work.
†Corresponding author.

39th Conference on Neural Information Processing Systems (NeurIPS 2025).

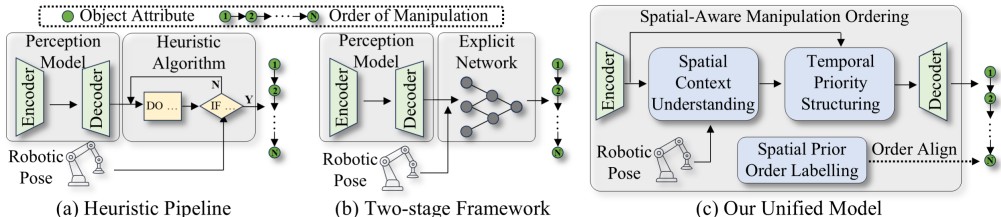

Figure 1: **Comparison of various designs for manipulation ordering.** (a) Heuristic pipelines use hand-crafted optimization algorithms for post-processing, but struggle with generalization. (b) Two-stage frameworks utilize separate networks to predict manipulation order, resulting in increased latency and reduced efficiency. (c) Our unified spatial-aware framework jointly learns spatial representation and manipulation ordering, achieving both high accuracy and real-time performance.

As illustrated in Fig. 1(c), we propose OrderMind, a unified framework that learns spatial-aware manipulation priorities directly from cluttered environments. It combines a spatial context understanding module and a temporal priority structuring module to infer manipulation ordering based on local geometry and interactions. A k-Nearest Neighbors algorithm-based spatial graph is proposed to capture object-object and object-manipulator relations, supporting efficient spatial layout understanding. We introduce a spatial prior labeling method that guides a vision-language model to generate plausible orderings. This enables training without manual labels. We modify a vision-centric backbone [11] for spatial context understanding. By integrating object understanding and order prediction into a single inference pass, the model achieves accurate and efficient manipulation ordering, maintaining spatial consistency across diverse cluttered scenes while supporting real-time performance.

To facilitate further assessment, we build the first large-scale simulation Manipulation Ordering Benchmark, which is explicitly designed for the manipulation ordering challenge and comprises 163,222 samples across diverse difficulty modes. We propose a set of evaluation metrics designed to thoroughly measure model performance, taking into account the validity of inferred manipulation sequences. Additionally, we conduct comprehensive experimental validation in both simulated and real-world settings. Our findings confirm the strength of the proposed approach, achieving real-time manipulation ordering in cluttered environments.

Our contributions are summarized as follows: (i) We propose a unified spatial-aware manipulation ordering framework for cluttered environments, which addresses the fundamental challenge of fully leveraging spatial information to learn manipulation ordering. (ii) To implement our proposed strategy, we introduce the *OrderMind* architecture, which consists of a spatial context understanding module and a temporal priority structuring module. This design enables efficient, real-time inference of manipulation order. (iii) Extensive experiments in both simulation and real-world robot setups demonstrate the effectiveness and robustness of our method across diverse cluttered scenes.

## 2 Related Works

### 2.1 Models for Robotic Manipulation

With diverse modeling paradigms ranging from vision-language reasoning to physics-based dynamics, recent models for robotic manipulation aim to bridge perception and control through both learned and analytical representations. Some vision-language models, such as CLIPort [12] and PaLM-E [13], extend manipulation capabilities by conditioning on language and integrating multimodal information from vision and proprioception. SayCan [14] introduced affordance-aware language grounding. RT-1 [15] and its successor RT-2 [16] re-frame actions as text tokens and co-finetune on internet-scale vision-language corpora, unlocking chain-of-thought reasoning for semantic and arithmetic object commands. VIMA [17] generalises prompt engineering to robotics for strong zero-shot performance.

ReKep [18] leveraged the power of VLM, specifically GPT-4o [19], to effectively reason about 3D rotations. It achieved this by expressing spatial relationships as arithmetic operations between keypoints. Moreover, advances in perception method provide scene reconstruction [20] and object understanding [21, 22] that can be used to preprocess visual inputs before manipulation. SeeDo [23] utilizes VLM with chain-of-thought prompting and visual cues from video tracking to enable effective

reasoning for robotic task planning. SpatialVLM [24] enhanced VLM spatial reasoning capabilities by training on a large-scale generated spatial Visual Question Answering dataset. Such perception-driven priors enhance spatial awareness [25, 26] and improve the reliability of downstream control. RoboMamba [27] integrated a vision encoder with Mamba [28] and Large Language Model to obtain both visual common sense and robotic-related reasoning abilities. Although VLM-based models possess general reasoning capability, they are relatively slow and cannot meet the real-time requirements of robotic manipulation tasks.

## 2.2 Manipulation Relationship Network

Early methods in visual relationship detection focused on building subject-predicate-object triplets and combining linguistic priors or global context for joint learning. Visual relationship prediction model [29] learned the appearance of objects and predicates individually and used the relationship embedding space learned from language to detect visual relationships. SCG [30] applied interactive message passing for deep object-relation interactions. VRL [31] used a reinforcement learning-based method to explore multiple relationships and attributes sequentially. CDDN Network [32] utilized word semantics and visual scene graphs to capture the interactions between different object instances for more accurate predictions. Graph R-CNN [33] leverages object-relationship regularities to generate scene graphs from detected objects in an end-to-end manner. Motifnet [34] staged detection results and relationships to establish a rich context for subsequent prediction.

To further integrate the physical order of actions into robotic manipulation. VMRN [35] proposed a novel pooling layer to predict manipulation relations in parallel. GVMRN [36] adopted GCNs to enhance contextual reasoning and filter irrelevant pairs, boosting inference robustness. MRG [37] used affordance representation to get the potential manipulation relationships of an arbitrary scene. VeriGraph[38] generated high-level task plans for robot object manipulation based on the scene graph. D3G[39] detected objects and reasoned relationships for grasp planning simultaneously by leveraging a transformer-based detector and graph layers. Although these models are capable of generating object relation graphs that can be used for manipulation tasks, they still focus on perceiving relationships between objects, ignoring the relationships between objects and the robot, and still require further post-processing to obtain the manipulation order.

# 3 Methodology

As illustrated in Fig. 2, we propose **OrderMind**, a unified framework for learning spatial-aware manipulation ordering in cluttered scenes. OrderMind integrates scene comprehension with manipulation ordering, enabling robots to perform real-time manipulation ordering in cluttered environments. This section describes how we define order-aware representation (Sec. 3.1), how to learn the spatial-aware manipulation ordering (Sec. 3.2), and how to construct reliable order annotations (Sec. 3.3).

## 3.1 Problem Formulation

We define the manipulation ordering challenge as learning a mapping $f : \mathcal{I} \times \mathrm{SE}(3) \to \mathcal{O}$, where $\mathcal{I} \subset \mathbb{R}^{H \times W \times 4}$ denotes the space of RGB-D images, $\mathrm{SE}(3)$ represents the pose space of the robotic end-effector, and $\mathcal{O} = \{\mathbf{O}, \Sigma\}$. The output $\mathcal{O}$ comprises two key components: a set of object representations $\mathbf{O} = \{\mathbf{o}_1, \mathbf{o}_2, \ldots, \mathbf{o}_n\}$, and a manipulation sequence $\Sigma = \{\sigma_1, \sigma_2, \ldots, \sigma_n\}$ that defines the operational priority order.

Each object representation $\mathbf{o}_i$ integrates both intrinsic attributes such as extent and category, and extrinsic attributes such as position and orientation, collectively capturing the spatial context necessary for reasoning about manipulation order. The model assigns each object a continuous priority score $s_i \in \mathbb{R}$ based on these spatial-aware representations, and the sequence $\Sigma$ is derived by sorting these scores. This approach enables fine-grained differentiation between manipulation priorities rather than simple ordinal rankings, particularly valuable when certain objects require significantly higher operational precedence.

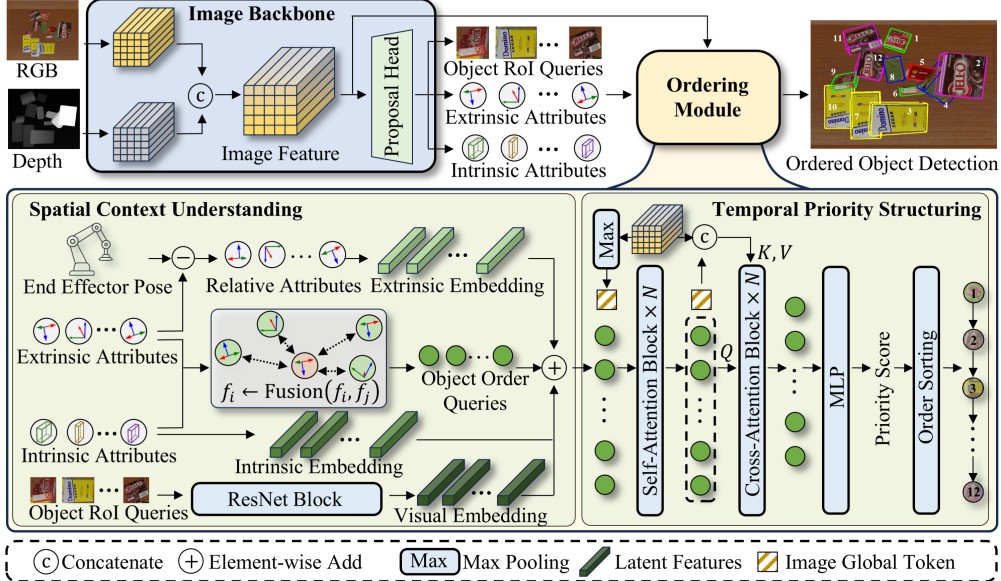

Figure 2: **Main framework of OrderMind.** OrderMind captures spatially-aware manipulation ordering representations that encode both object-object relationships and object-manipulator interactions in cluttered environments. The spatial context understanding module aggregates spatial information from local geometry, while the temporal priority structuring module integrates local object features with global scene context. OrderMind predicts the manipulation priority for each object, and by sorting these priorities, we can obtain the manipulation order.

## 3.2 Spatial-Aware Manipulation Ordering

To support optimal prediction of manipulation order, we design both a spatial context understanding module and a temporal priority structuring module, which tightly integrate spatial-aware arrangements and manipulation ordering.

**Spatial Context Understanding.** Occlusions and inter-object support in cluttered scenes make it necessary to reason about spatial and physical relationships rather than visual appearance alone. To enable accurate manipulation ordering in cluttered environments, we focus on extracting spatial-aware representations that capture both local and global scene structure. We represent each object using its 3D bounding box center and associated intrinsic and extrinsic attributes. The intrinsic attributes describe inherent properties such as semantic class and physical extent, while the extrinsic ones define the object's pose in the world coordinate system. The object centers collectively form a sparse point cloud representing the 3D scene layout. Based on this point cloud, we construct a spatial graph where nodes correspond to objects and edges encode geometric proximity.

To model inter-object spatial dependencies, we apply a k-Nearest Neighbors strategy to build a localized spatial graph around each object. As shown in Fig. 3, we treat the center point of each object as a node in the graph and connect each object to its neighbors with edges. Each node in this graph incorporates both intrinsic and extrinsic attributes. For each center $p_i$, messages from its spatial neighbors $\mathcal{N}_k(p_i)$ are aggregated to form spatial-aware features that encode local geometric structure, as described in Equation 1:

$$\text{Fusion}\left(f_i, f_j\right) = \mathcal{M}(\text{Linear}(\text{Concate}(f_i, f_j - f_i))), \quad \forall p_j \in \mathcal{N}_k(p_i), \tag{1}$$

This design captures how physical proximity and spatial alignment influence the feasibility of manipulation actions. In line with PointNet-style architectures [40, 41], feature aggregation is performed using max pooling $\mathcal{M}$ across neighborhoods to produce compact object-level embeddings.

In addition to object-object interactions, we model the spatial relationship between each object and the robot. This is encoded by computing the relative transformation between each object's pose and the end-effector's current state, providing important cues for operational accessibility. These robot-centric features are integrated into the object representation to further refine the spatial-aware

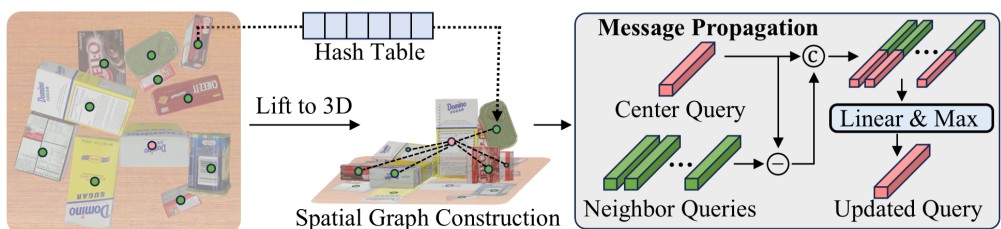

Figure 3: **Details of Spatial Context Understanding Module.** A spatial graph is constructed from object-centric point clouds to capture scene layout in cluttered environments. Local geometric structure is extracted using a k-Nearest Neighbors strategy, and spatial-aware features are aggregated through neighborhood pooling to support manipulation ordering.

embedding. By jointly encoding object geometry, visual cues, and robot-relative spatial structure, our method constructs a rich spatial-aware representation that supports effective manipulation ordering in cluttered environments.

**Temporal Priority Structuring.** Once spatial-aware features are extracted, the model transitions from structural perception to manipulation-oriented ordering by learning to prioritize objects according to their contextual importance. To this end, we design an attention mechanism to dynamically integrate object-specific information with scene-level cues.

Firstly, a set of object ordering tokens extracted from the image encoder is aggregated via a global max pooling to form a high-level scene representation, denoted as $G$. This global context encodes the layout of the scene, capturing occlusion patterns, object stacking, and spatial symmetry. Each object token $Q$ is then refined through self-attention, which incorporates inter-object interactions and aligns the object tokens with one another. Formally, the attention flow is defined as:

$$[Q', G] \leftarrow \text{Self-Attn}([Q, G]), \qquad Q'' \leftarrow \text{Cross-Attn}(Q', [G, F], [G, F]), \qquad (2)$$

where $[\cdot, \cdot]$ denotes the concatenate operation, $F$ denotes the original visual features and $Q'$ represents updated object embeddings after intra-token refinement. Secondly, the cross-attention module allows each object token to query both the global context $G$ and its associated visual representation $F$, enabling the model to extract manipulation-relevant cues while maintaining spatial coherence. Finally, the ordering token are decoded into a set of priority scores, which represent the manipulation priority for each object under the current spatial layout.

The output $Q''$ encodes spatial-aware priority for each object, implicitly modeling manipulation priority under geometric and physical constraints. By leveraging structured attention, our method enforces consistency between local feasibility and global accessibility, resulting in manipulation orderings that are both spatially grounded and temporally executable.

**Preferential Order Alignment.** To learn a spatially-aware manipulation ordering from discrete order annotations, we first align predicted and ground-truth objects via bipartite matching. Let $o$ and $\hat{o}$ denote ground-truth and predicted objects, respectively. We formulate the assignment using the Hungarian algorithm, where object similarity serves as the matching cost $\hat{\sigma} = \arg\min_{\sigma} \sum_{i=1}^{N} \mathcal{L}_{\text{hungarian}}\left(o_i, \hat{o}_{\sigma(i)}\right)$. When the object correspondence $\hat{\sigma}$ established, we align the prediction of ordering preference with ground truth order by comparing pairwise relations. Instead of directly predicting discrete ranks, the model learns to produce continuous scores $\hat{s}$ to implicitly align sorted order with the annotated order. The objective is expressed as follows:

$$\mathcal{L}_{\text{order}} = \sum_{j=1}^{N} \sum_{k=1}^{N} w_{jk} \cdot \mathbb{1}_{\{o_j < o_k\}} \log\left(1 + \exp\left(\hat{s}_{\hat{\sigma}(k)} - \hat{s}_{\hat{\sigma}(j)}\right)\right), \qquad (3)$$

where $w_{jk} = \log(1 + |o_j - o_k|)$ is a logarithmic weighting term that emphasizes ordering consistency for object pairs with large ground-truth rank differences.

This formulation allows the model to infer object priority through relative comparisons in the score space, rather than committing to exact rank values. The transition from discrete index annotations to continuous score predictions enables the model to capture more reasonable preferences in manipulation order, especially in cases where certain objects dominate the ordering due to spatial accessibility

or structural dependencies. By sorting these predicted scores, the final output ordering naturally aligns with spatial constraints and task demands.

## 3.3 Spatial Priors Order Labeling

Learning spatial-aware manipulation ordering relies on well-structured annotations that reflect object interaction priorities in cluttered environments. In our framework, these manipulation orders are generated by a VLM [42]. To enhance the quality and consistency of these labels, we introduce two spatial priors to serve as auxiliary signals during the generation process. These priors influence how the large model interprets spatial relationships, ensuring that the predicted orderings align with common physical and operational patterns observed in cluttered scenes.

**Independence Prior.** This prior encourages the early manipulation of objects that are spatially separated from others on the horizontal plane. In cluttered environments, such independent objects can be accessed more safely, as their removal is unlikely to cause physical interference with neighboring items. For each object $\mathbf{o}_i \in \mathbf{O}$, we compute its projected area $A_i$ by projecting its 3D bounding box onto the horizontal support surface. The pairwise distance between two objects is defined as $d(A_i, A_j) = \inf\{\, \|p_i - p_j\|_2 \mid p_i \in A_i,\ p_j \in A_j \,\}$. An object is considered spatially independent when the minimum distance from $A_i$ to all other $A_j$ is no less than a threshold $\tau$, which is selected based on the size of the suction end-effector and a required safety margin:

$$I_{\text{indep}}(\mathbf{o}_i, \mathbf{O}) = \begin{cases} 1, & \text{if } \min_{j \neq i} d(A_i, A_j) \geq \tau \\ 0, & \text{otherwise} \end{cases}, \tag{4}$$

where objects satisfying this condition are generally safe to manipulate first, providing useful spatial cues for generating physically consistent manipulation order labels.

**Local Optimality Prior.** This prior captures vertical accessibility by identifying objects that are not occluded from above. In cluttered environments, vertical stacking is frequent, and selecting objects from the top layers helps avoid disturbing other items and maintains the scene's stability. For object $\mathbf{o}_i \in \mathbf{O}$ and its 3D bounding box $B_i$, we define $z_{\max}(\mathbf{o}_i) = \sup\{z \mid (x, y, z) \in B_i\}$ as the height of its topmost point. The volume above $\mathbf{o}_i$ is defined as the upward extension of its projected area $A_i$:

$$V_{\text{above}}(\mathbf{o}_i) = \left\{ (x, y, z) \in \mathbb{R}^3 \mid (x, y) \in A_i \wedge z > z_{\max}(\mathbf{o}_i) \right\}. \tag{5}$$

An object is locally optimal when no other object intersects with this vertical space:

$$I_{\text{topmost}}(\mathbf{o}_i, \mathbf{O}) = \begin{cases} 1, & \text{if } \forall k \neq i, B_k \cap V_{\text{above}}(\mathbf{o}_i) = \emptyset \\ 0, & \text{otherwise} \end{cases}, \tag{6}$$

where objects that satisfy this criterion are likely to be directly accessible from above. This prior provides useful spatial context to help the model generate orders that respect scene stability and reduce structural disruption during manipulation.

## 4 Experiments

This section evaluates our framework for its effectiveness in manipulating ordering in cluttered scenes. Section 4.1 provides the details related to the image datasets and implementation. Section 4.2 presents the results obtained with our framework and compares them with other related approaches. Finally, section 4.3 contains ablation studies to investigate the effectiveness of each proposed component.

### 4.1 Experimental Settings

#### 4.1.1 Environments and Task

We constructed datasets involving complex robotic manipulation scenarios in both simulated and real-world environments. For simulation, we utilized Pybullet [43] as the engine and designed cluttered scenes using the YCB object set [44]. Three levels of difficulty were established: Easy with 24 objects, Medium with 36 objects, and Hard with 60 objects. The simulation dataset consists of 161,722 RGB-D images for training, 1,500 images for validation. Closed-loop testings are performed within the PyBullet simulator. For real-world experiments, we collected 26,324 images for training and 6,581 for validation. Closed-loop testings are directly conducted in real-world cluttered environments.

Table 1: **Manipulation performance** under easy, moderate, and hard settings in the simulation environment. RC, OD and SR refer to residual count, object disturbance and success rate, respectively. Privilege indicates models are accessible to objects' ground truth information. N/A indicates that the model's parameter size is not reported.

| Framework | Privilege | Easy | | | Moderate | | | Hard | | | #Param. | FPS |
|---|---|---|---|---|---|---|---|---|---|---|---|---|
| | | RC ↓ | OD ↓ | SR (%) ↑ | RC ↓ | OD ↓ | SR (%) ↑ | RC ↓ | OD ↓ | SR (%) ↑ | | |
| GPT-4o [19] | ✓ | 4.8±1.0 | 2.3±0.8 | 90.3±6.4 | 8.3±1.3 | 5.0±1.0 | 77.9±8.4 | 16.8±2.2 | 12.0±2.6 | 71.4±8.3 | N/A | 0.1 |
| Claude-3.7-Sonnet [49] | ✓ | 0.7±0.6 | 2.0±0.4 | 92.1±7.1 | 2.2±1.1 | 4.5±1.1 | 84.0±7.1 | 5.3±2.2 | 10.4±2.5 | 77.9±8.3 | N/A | 0.1 |
| Gemini-2.5 [50] | ✓ | 0.7±0.9 | 1.6±0.7 | 92.4±9.7 | 2.2±1.8 | 4.8±1.5 | 84.6±10.8 | 5.8±6.1 | 12.4±5.0 | 78.5±16.5 | N/A | 0.1 |
| Gemma-3 [51] | ✓ | 1.2±1.2 | 2.2±1.2 | 88.3±12.5 | 2.4±1.7 | 4.9±1.8 | 83.2±10.3 | 7.5±3.2 | 11.2±2.2 | 70.6±9.3 | 27B | 0.1 |
| Qwen2.5-VL [42] | ✓ | 0.9±0.9 | 1.8±0.8 | 92.5±10.3 | 2.7±1.5 | 4.5±1.1 | 82.1±9.4 | 7.7±3.7 | 11.4±3.4 | 70.4±11.6 | 72B | 0.1 |
| InternVL3 [52] | ✓ | 1.4 ±1.0 | 2.3±0.8 | 88.2±10.9 | 2.8±1.9 | 4.8±2.2 | 82.1±11.7 | 12.1±14.2 | 13.9±4.5 | 70.4±8.9 | 14B | 0.2 |
| YOLOv11-seg [53]+SPH | ✗ | 4.7±1.1 | 1.6±0.6 | 75.1±4.5 | 6.0±1.5 | 4.0±0.8 | 77.5±6.4 | 9.8±3.1 | 8.7±2.0 | 76.4±6.0 | 31.7M | 11.9 |
| YOLOv11-det [53]+SPH | ✗ | 3.4±2.1 | 2.0±0.5 | 75.5±8.4 | 4.6±1.4 | 3.9±0.9 | 79.8±7.0 | 10.3±2.5 | 8.2±0.9 | 74.9±6.2 | 31.7M | 11.9 |
| UniDet3D [54]+Confidence | ✗ | 3.1±1.6 | 2.5±0.8 | 49.4±17.0 | 4.5±2.3 | 4.8±1.5 | 53.0±7.9 | 9.7±2.5 | 8.4±1.6 | 46.1±10.9 | 15.8M | 0.8 |
| UniDet3D+Distance | ✗ | 2.3±1.0 | 2.4±0.7 | 40.3±10.0 | 4.9±1.9 | 5.5±1.5 | 34.8±7.8 | 14.5±3.6 | 12.8±3.9 | 24.9±3.8 | 15.8M | 0.8 |
| UniDet3D+GPT-4o | ✗ | 7.2±1.6 | 2.0±0.8 | 42.4±7.2 | 12.0±2.5 | 3.9±0.9 | 39.6±7.9 | 23.6±3.4 | 9.2±1.2 | 33.4±5.6 | 15M+N/A | 0.1 |
| UniDet3D+Claude-3.7-Sonnet | ✗ | 4.7±4.6 | 2.8±1.1 | 39.9±6.7 | 6.5±2.9 | 6.1±0.9 | 36.0±11.9 | 18.6±4.0 | 14.2±1.5 | 30.7±5.7 | 15M+N/A | 0.1 |
| UniDet3D+Gemini-2.5 | ✗ | 3.2±2.0 | 3.0±1.2 | 45.5±20.3 | 5.6±3.0 | 5.5±1.8 | 46.4±11.2 | 16.5±3.7 | 13.1±2.1 | 26.6±2.0 | 15M+N/A | 0.1 |
| UniDet3D+Gemma-3 | ✗ | 3.1±0.9 | 2.8±0.9 | 40.3±15.0 | 5.5±2.3 | 5.3±1.4 | 42.2±6.7 | 15.3±3.6 | 13.5±3.6 | 35.8±5.9 | 15M+27B | 0.1 |
| UniDet3D+Qwen2.5-VL | ✗ | 3.5±2.2 | 3.4±1.1 | 38.8±14.3 | 7.7±2.6 | 7.1±1.7 | 34.5±11.7 | 19.9±10.0 | 13.2±2.3 | 28.3±5.9 | 15M+72B | 0.1 |
| UniDet3D+InternVL3 | ✗ | 2.8 ±1.2 | 2.9±1.1 | 38.3±10.1 | 7.1±1.7 | 5.8±1.5 | 42.9±8.5 | 16.8±6.1 | 11.4±1.3 | 43.1±9.4 | 15M+14B | 0.2 |
| OrderMind-Mini (Ours) | ✗ | 1.0±1.0 | 1.6±0.8 | 94.2±7.6 | 3.3±1.6 | 3.6±1.0 | 89.6±7.1 | 5.4±1.3 | 7.2±1.4 | 90.4±3.5 | 35.2M | **21.3** |
| OrderMind (Ours) | ✗ | **0.4±0.5** | **0.7±0.3** | **96.5±5.3** | **1.0±0.9** | **1.4±0.5** | **95.3±5.1** | **3.3±1.4** | **4.3±1.5** | **95.4±5.1** | **41.8M** | 5.6 |

The task is to pick up objects sequentially in scenes with densely stacked items. It requires maintaining a high success rate while minimizing disturbance to surrounding objects. OrderMind replans after each manipulation to ensure robustness against disturbance from object interactions. The robot interacts with objects using a suction-based end-effector. An RGB-D camera, positioned in a top-down orientation above the workspace, captures visual input at a resolution of $1408 \times 1024$.

### 4.1.2 Training and Implementation Details

We trained our model on a single RTX 4090 GPU using a batch size of 24. The optimizer is AdamW [45] with an initial learning rate of $2 \times 10^{-4}$ and a weight decay of 0.01. The learning rate follows a cosine annealing schedule [46]. Training proceeds in two stages. The first stage focuses on pre-training the image backbone [47, 48] for 50 epochs. In the second stage, the full model is trained end-to-end for 10 epochs, enabling the system to learn order-aware manipulation directly from sufficient spatial representation.

### 4.1.3 Evaluation Metrics

We evaluate performance using three metrics: higher success rates, lower object disturbance, and fewer residual objects, which reflect a more effective manipulation order.

**Success Rate.** This metric is defined as the ratio of successful pick-ups to total attempts. In simulation, a pick is considered successful when the 6D pose of the end-effector $\mathbf{p}_e$, including both its center and orientation, closely aligns with the target object pose $\mathbf{p}_{obj}$. Specifically, success is confirmed when the positional difference satisfies $\|\mathbf{p}_e - \mathbf{p}_{obj}\|_2 < 0.05$.

**Residual Count.** This metric tracks the number of objects left outside the robot's reachable workspace at the end of a task. A smaller count indicates that the robot maintained optimal manipulation ordering during the task. While all objects start within reach, due to unintended displacement during manipulation, some may shift out of range and remain unprocessed.

**Object Disturbance.** In cluttered environments, manipulating one object can cause disturbance to nearby objects. After each action, the total displacement of surrounding objects is measured. Lower disturbance indicates that a well-ordered manipulation sequence is being operated by the robot, ensuring the scene remains stable during each step of order-aware manipulation.

## 4.2 Experimental Results

### 4.2.1 Simulation Experiments

Tab. 1 presents a comprehensive evaluation of manipulation performance in cluttered environments across four experimental settings: heuristic pipeline, two-stage framework, privilege information-based two-stage framework, and our proposed unified spatial-aware framework. Privilege information

Table 2: Comparison of order stability and success rate under different replanning intervals. Order-Mind maintains lower Levenshtein Distance (LD) and higher Success Rate (SR).

| Interval | Framework | Privilege | LD↓ | SR↑ |
|---|---|---|---|---|
| 1 | GT+SPH | ✓ | 1.5 | 96.3 |
| | YOLO11-seg+SPH | ✗ | 3.6 | 77.5 |
| | YOLO11-det+SPH | ✗ | 1.8 | 79.8 |
| | OrderMind (Ours) | ✗ | 1.7 | 89.6 |
| 5 | GT + SPH | ✓ | 5.2 | 85.5 |
| | YOLO11-seg+SPH | ✗ | 6.6 | 75.3 |
| | YOLO11-det+SPH | ✗ | 6.5 | 72.2 |
| | OrderMind (Ours) | ✗ | 3.5 | 82.3 |
| 10 | GT + SPH | ✓ | 7.5 | 85.5 |
| | YOLO11-seg+SPH | ✗ | 9.3 | 70.6 |
| | YOLO11-det+SPH | ✗ | 9.3 | 69.0 |
| | OrderMind (Ours) | ✗ | 4.7 | 80.5 |

refers to models having direct access to ground truth object pose information in the simulation environment. We derive our Spatial Priors (Sec. 3.3) as the Spatial Priors Heuristic (SPH) for complete comparison. The speed of VLMs is calculated by using OpenRouter [55].

We first evaluate the upper bound of vision-language reasoning using VLMs with privileged information. Even with privileged information, the best model achieves only 92.5% success rate in easy setting and 78.5% in hard setting. Next, we test heuristic pipelines using YOLOv11 [53] and UniDet3D [54], selecting objects by Spatial Priors Heuristic, confidence, and distance to the end effector. Although UniDet3D and YOLO has fewer parameter counts, it is limited to 0.8 FPS due to its point cloud-based architecture. We also evaluate two-stage frameworks where UniDet3D results are passed to VLMs for ordering, which achieves only 0.1 FPS and cannot meet the real-time demand.

Our OrderMind framework learns order-aware manipulation directly from raw visual inputs, enabling precise sequence planning and stable performance in cluttered environments. Its core strength is its spatial understanding, which anticipates cascading effects to minimize disturbance to surrounding objects. OrderMind yields superior performance as it achieves a 96.5% success rate in easy mode, outperforming even privileged VLMs, and maintains a 95.4% success rate in hard mode. In contrast, Gemini-2.5's performance drops to 78.5%, which demonstrates that conventional VLMs struggle to infer stacking relations and plan effective sequences. Ultimately, OrderMind's ability to learn spatial order and minimize disruption provides a clear advantage in cluttered robotic tasks. Furthermore, the lightweight OrderMind-Mini model delivers superior accuracy in real-time (21.3 FPS) with only 35.2M parameters, confirming the framework's strong potential for real-world deployment.

**Order Stability Analysis.** To examine the effect of replanning frequency on plan stability and task performance, we compare OrderMind against heuristic baselines under various replanning intervals. We use the Levenshtein Distance (LD) to quantify plan stability, where a larger LD indicates greater order reshuffling between consecutive plans.

As shown in Tab. 2, our method delivers more stable plans and a higher success rate compared to heuristic baselines. With infrequent replanning, these baselines produce large order reshuffling reflected by a larger LD, and suffer a drop in success. However, our approach consistently outperforms the two-stage framework in all tested intervals, achieving both a lower LD and a superior success rate.

We attribute this superior performance to our unified design that jointly learns spatial representations and manipulation order. Heuristic methods are often greedy, i.e., they select the best immediate action but fail to foresee long-term consequences. In contrast, OrderMind distills knowledge from VLMs and integrates rich visual and geometric context. This allows it to generate globally coherent and far-sighted plans, making them inherently more stable and successful.

**Robustness under Simulated Label Noise.** To assess our OrderMind's robustness to unexpected noise from VLMs such as hallucinated spatial relations, incorrect labels, or omissions, we conducted a study by randomly perturbing a certain percentage of pairwise orders in the VLM labels.

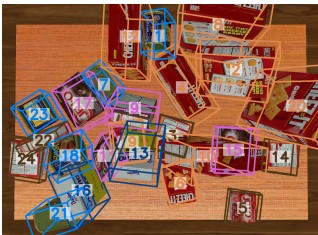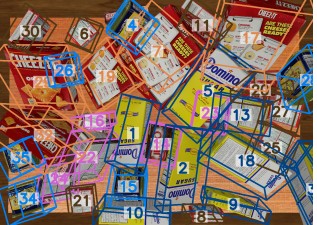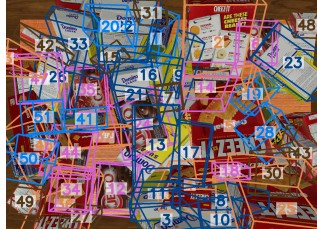

Figure 4: **Visualization of manipulation ordering results** under easy, moderate, and hard settings. 3D bounding boxes are demonstrated and manipulation orders are marked at the center of each object.

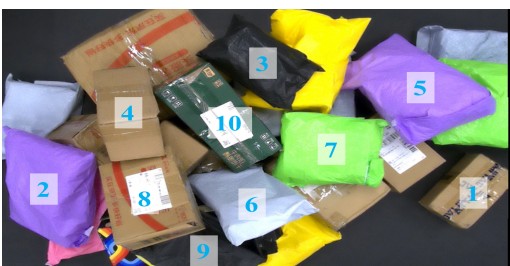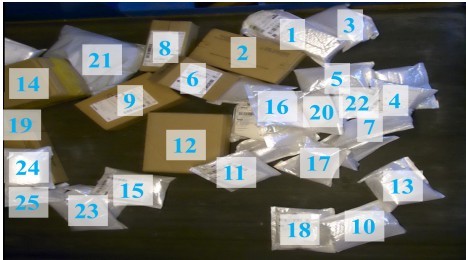

Figure 5: **Visualization of real-world manipulation ordering.** The left figure shows results from experiments conducted in a laboratory setting, while the right figure presents results obtained in an actual factory environment. Predicted order is annotated on each object in the scene.

Tab. 3 reports success rates on scenes of varying difficulty under different noise ratios. The performance degrades slightly with moderate noise, demonstrating the robustness of our method to VLM label noise, thanks to both the label ensemble strategy and the unified learning of perception and ordering. However, under the extreme 70% noise condition, the performance drops substantially. This likely occurs because, at such a high noise ratio, the model fails to extract meaningful information and effectively degenerates into a random policy. In moderate mode, the success rate also declines due to perception noise–induced errors.

Table 3: Comparison of success rates under varying levels of simulated label noise.

| Noise Ratio | Privilege | SR-Easy↑ | SR-Mod↑ | SR-Hard↑ |
|---|---|---|---|---|
| 0 | ✗ | 85.35 | 83.19 | 83.66 |
| 0.1 | ✗ | 85.10 | 82.71 | 79.13 |
| 0.2 | ✗ | 80.08 | 79.00 | 76.30 |
| 0.5 | ✗ | 78.88 | 76.91 | 73.81 |
| 0.7 | ✗ | 75.99 | 70.36 | 67.31 |
| GT+random | ✓ | 75.50 | 75.58 | 68.88 |

#### 4.2.2 Real World Experiments

Tab. 4 shows the performance of our OrderMind model in real-world experiments across three levels of task difficulty. As the task difficulty increases from easy to hard, the success rate decreases from 93.3% to 76.6%, while the residual count rises from 0.2 to 3.0. This reflects the growing challenge of planning and executing actions in denser and more constrained object arrangements. Despite this, the model continues to demonstrate a strong ability to complete tasks with relatively high success rates and limited residual objects.

Table 4: Performance of Order-Mind in real-world closed-loop testing under three difficulty modes.

| Mode | RC | SR (%) |
|---|---|---|
| Easy | 0.2 | 93.3 |
| Moderate | 2.0 | 78.5 |
| Hard | 3.0 | 76.6 |

Fig. 5 illustrates the predicted manipulation sequences in real-world scenes, with numbers directly annotated on the objects to indicate their execution order. The results show that our model can effectively understand both object isolation and stacking relationships. Guided by the learned sequential policy, the robot interacts with objects efficiently and safely, while preserving scene stability. This capability is essential for enabling reliable and autonomous robotic operation in real-world environments.

**Failure Analysis.** Reviewing 30 minutes of robotic arm operation, we found four main failure sources. Perception errors were primary contributor, with 21% from inaccurate 3D rotation and another 15% from object center point misidentification. For challenges with deformable objects, an inability to find suitable suction surfaces caused 21% of the failures. Furthermore, incorrect manipulation ordering led to object disturbances, accounting for 39%. Other factors, like poor synergy between the robotic arm and the camera, accounted for 4%.

### 4.3 Ablation Study

We conduct an ablation study to assess the contribution of each core component in our framework, as demonstrated in Tab. 5. Starting from a baseline with 5.0 residual objects, 5.4 total displacement and a success rate of 76.1%, we observe steady improvements with the addition of each module.

Introducing either SCU or TPS individually boosts the success rate to 81.0% and 80.5%, while also reducing residual objects and disturbance. Their combination yields further gains, with success rate reaching 87.7%, suggesting that accurate feature association and context-aware selection complement each other in improving manipulation quality. Adding the SPOL component leads to the most substantial improvements. For example, the combination of SCU and TPS lifts the success rate to 91.4% and reduces the number of residual objects to 2.0, demonstrating the importance of learning informed action sequences in cluttered environments.

Table 5: Ablation study of proposed modules on moderate task level. SCU, TPS, SPOL denote spatial context understanding, temporal priority structuring, spatial priors order labeling, respectively.

| SCU | TPS | SPOL | RC ↓ | OD ↓ | SR(%) ↑ |
|---|---|---|---|---|---|
| | | | 5.0 | 5.4 | 76.1 |
| ✓ | | | 3.6 | 4.4 | 81.0 |
| | ✓ | | 4.5 | 4.9 | 80.5 |
| ✓ | ✓ | | 3.5 | 4.4 | 87.7 |
| ✓ | | ✓ | 2.0 | 4.0 | 91.4 |
| | ✓ | ✓ | 3.4 | 3.9 | 90.5 |
| ✓ | ✓ | ✓ | **1.0** | **1.4** | **95.3** |

Finally, the full model with SCU, TPS, and SPOL achieves the best performance with only 1.0 residual object, and minimal disturbance at 1.4. These results highlight the strong synergy among components—feature association, object selection, and order planning together lead to efficient and stable manipulation.

## 5 Conclusion and Limitation

In this work, we presented **OrderMind**, a framework for learning spatial-aware manipulation ordering tailored to robotic tasks in cluttered environments. OrderMind addresses this by integrating spatial context understanding and temporal priority structuring, enabling reliable and efficient manipulation without reliance on idealized inputs or environment models. We demonstrate that OrderMind achieves high success rates, minimal object disturbance, and low residuals across both simulated and real-world cluttered environments, all while operating in real time. We found that while large models provide strong general scene understanding and ordering capabilities, their reasoning remains limited when confronted with the fine-grained spatial dependencies and object stacking present in cluttered settings.

**Broader Impact.** OrderMind can improve the reliability and autonomy of robots in cluttered environments, enhancing efficiency and safety in sorting, warehouse logistics, and assistive robotics. It can reduce manipulation errors, making automation more sustainable and effective. The spatial-aware encoding ability also enables easier deployment in unstructured settings, broadening access to intelligent automation in healthcare, manufacturing, and services, while inspiring new research in human-robot collaboration.

**Limitations**. Despite these advantages, several limitations remain. The current system assumes a stable scene during execution, which may restrict its ability to adapt to dynamic environments. Besides, the accuracy of order-aware manipulation prediction relies on precise 3D attribute estimation, which remains challenging under severe occlusions common in cluttered environments.

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

# A    Dataset Analysis

This section details the object categories featured in our dataset. In the simulated environments, we incorporated five categories derived from the YCB dataset [44], namely CrackerBox, GelatinBox, PottedMeatCan, SugarBox, and PuddingBox. For our real-world dataset, we employed the two broader categories of box and bag. Objects classified as boxes generally possess a firm shape, facilitating perception. Conversely, bag objects are prone to deformation, thereby posing a greater challenge to accurate spatial layout perception.

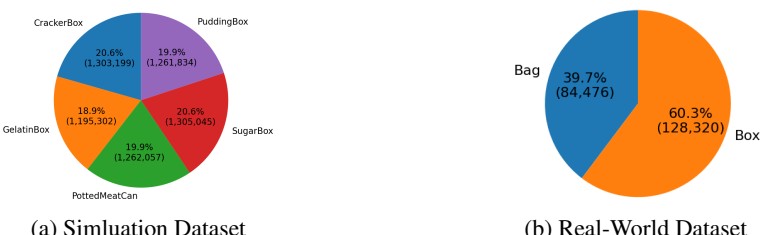

(a) Simluation Dataset          (b) Real-World Dataset

Figure 6: Visualization of the dataset category distribution pie chart, with object counts shown in parentheses.

# B    Data Annotation Pipeline

This section details our methodology for generating spatial layout perception labels within real-world environments. To elaborate, for a given RGB image, we initially employ Segment Anything [56] to produce instance-level segmentation masks. These masks are subsequently integrated with corresponding RGB-D data to create instance-level point clouds. We employ the depth inpainting method [57] to reduce noise during the data annotation stage. Following this, the RANSAC algorithm [58] is utilized to extract the dominant planar surface from these point clouds.

# C    Additional Experimental Results

## C.1    Comparison with Baselines under Multiple Difficulty Mode

In this section, we present additional experimental result to demonstrate the effectiveness of our OrderMind model architecture. Tab. 6, Tab. 7 and Tab. 8 are manipulation ordering results under easy, moderate and hard difficulty mode separately. Succ. Num., Fail Num., RC, OD and SR refer to number of successfully picked-up objects, number of failed pick-up attempts, residual count, object disturbance and success rate respectively. Privilege indicates models are accessible to objects' ground truth information.

These tables demonstrate that our OrderMind method achieves the best performance across easy, moderate, and hard difficulty levels. It is also superior in performance and faster than the VLM method with privilege information. Notably, OrderMind's performance advantage becomes more pronounced as the difficulty increases. Importantly, 'fail num' indicates the number of failed attempts. Thus, inaccurate detection results can lead the robotic end-effector to repeatedly try to pick false-positive objects, resulting in a higher num of failure attempts. So that experiments relying on UniDet3D [54] usually have a higher 'fail num' metric. A lower residual count, lower object disturbance, and higher success rate demonstrate OrderMind's strong understanding of scene stability and its ability to perform object manipulation quickly and accurately.

## C.2    Perception Result

The manipulation ordering task demands a robust understanding of spatial layout. To evaluate this spatial comprehension, we utilize detection tasks. Tab. 9 presents the detection mean Average Precision (mAP) under easy, moderate, and hard difficulty settings. While [59] suggests that point cloud-based methods generally achieve better performance in robotic tasks, our OrderMind method's superior mAP compared to such methods demonstrates that its end-to-end architecture mutually enhances both spatial layout perception and ordering capabilities.

## C.3    Generalization

To evaluate generalization beyond the YCB dataset, additional experiments were conducted in a novel and more challenging environment. The new setup replaces the flat tabletop with a deep-sided metal tray, introducing

Table 6: Manipulation performance under **easy** setting in the simulation environment. Succ. Num., Fail Num., RC, OD and SR refer to number of successfully picked-up objects, number of failed pick-up attempts, residual count, object disturbance and success rate, respectively. Privilege indicates models are accessible to objects' ground truth information.

| Framework | Privilege | Succ. Num.↑ | Fail Num.↓ | RC↓ | OD↓ | SR(%)↑ |
|---|---|---|---|---|---|---|
| GPT-4o [19] | ✓ | 19.2 | 2.1 | 4.8 | 2.3 | 90.3 |
| Claude-3.7-Sonnet [49] | ✓ | 23.3 | 2.1 | 0.7 | 2.0 | 92.1 |
| Gemini-2.5 [50] | ✓ | 23.3 | 2.1 | 0.7 | 1.6 | 92.4 |
| Gemma-3 [51] | ✓ | 22.8 | 3.4 | 1.2 | 2.2 | 88.3 |
| InternVL3 [52] | ✓ | 22.6 | 3.3 | 1.4 | 2.3 | 88.2 |
| Qwen2.5-VL [42] | ✓ | 23.1 | 2.1 | 0.9 | 1.8 | 92.5 |
| UniDet3D [54]+Confidence | ✗ | 20.9 | 26.4 | 3.1 | 2.5 | 49.4 |
| UniDet3D+Distance | ✗ | 21.7 | 36.1 | 2.3 | 2.4 | 40.3 |
| UniDet3D+GPT-4o | ✗ | 16.6 | 22.8 | 7.2 | 2.0 | 42.4 |
| UniDet3D+Claude-3.7-Sonnet | ✗ | 19.3 | 29.6 | 4.7 | 2.8 | 39.9 |
| UniDet3D+Gemini-2.5 | ✗ | 20.8 | 37.5 | 3.2 | 3.0 | 45.5 |
| UniDet3D+Gemma-3 | ✗ | 20.9 | 43.8 | 3.1 | 2.8 | 40.3 |
| UniDet3D+InternVL3 | ✗ | 21.2 | 36.0 | 2.8 | 2.9 | 38.3 |
| UniDet3D+Qwen2.5-VL | ✗ | 20.5 | 39.0 | 3.5 | 3.4 | 38.8 |
| OrderMind-Mini (Ours) | ✗ | 23.0 | 1.5 | 1.0 | 1.6 | 94.2 |
| OrderMind (Ours) | ✗ | 23.6 | 0.9 | 0.4 | 0.7 | 96.5 |

Table 7: Manipulation performance under **moderate** setting in the simulation environment. Succ. Num., Fail Num., RC, OD and SR refer to number of successfully picked-up objects, number of failed pick-up attempts, residual count, object disturbance and success rate, respectively. Privilege indicates models are accessible to objects' ground truth information.

| Framework | Privilege | Succ. Num.↑ | Fail Num.↓ | RC↓ | OD↓ | SR(%)↑ |
|---|---|---|---|---|---|---|
| GPT-4o | ✓ | 27.7 | 8.1 | 8.3 | 5.0 | 77.9 |
| Claude-3.7-Sonnet | ✓ | 33.8 | 6.6 | 2.2 | 4.5 | 84.0 |
| Gemini-2.5 | ✓ | 33.8 | 6.6 | 2.2 | 4.8 | 84.6 |
| Gemma-3 | ✓ | 33.6 | 7.2 | 2.4 | 4.9 | 83.2 |
| InternVL3 | ✓ | 33.2 | 7.8 | 2.8 | 4.8 | 82.1 |
| Qwen2.5-VL | ✓ | 33.3 | 7.6 | 2.7 | 4.5 | 82.1 |
| UniDet3D+Confidence | ✗ | 31.3 | 29.4 | 4.5 | 4.8 | 53.0 |
| UniDet3D+Distance | ✗ | 31.1 | 62.5 | 4.9 | 5.5 | 34.8 |
| UniDet3D+GPT-4o | ✗ | 24.0 | 37.6 | 12.0 | 3.9 | 39.6 |
| UniDet3D+Claude-3.7-Sonnet | ✗ | 29.5 | 69.9 | 6.5 | 6.1 | 36.0 |
| UniDet3D+Gemini-2.5 | ✗ | 30.4 | 37.6 | 5.6 | 5.5 | 46.4 |
| UniDet3D+Gemma-3 | ✗ | 30.5 | 43.3 | 5.5 | 5.3 | 42.2 |
| UniDet3D+InternVL3 | ✗ | 28.9 | 40.1 | 7.1 | 5.8 | 42.9 |
| UniDet3D+Qwen2.5-VL | ✗ | 28.3 | 59.1 | 7.7 | 7.1 | 34.5 |
| OrderMind-Mini (Ours) | ✗ | 32.7 | 3.9 | 3.3 | 3.6 | 89.6 |
| OrderMind (Ours) | ✗ | 35.0 | 1.8 | 1.0 | 1.4 | 95.3 |

distinct visual textures, backgrounds, and physical constraints for manipulation. A completely new set of five objects with diverse geometries and categories was used, including a handled laundry bottle, a thin correction-fluid pen, a cube-shaped tea box, a cylindrical conditioner bottle, and a rectangular book. Each object was randomly scaled by a random factor between 0.8 and 1.2 to further increase variability.

Although UniDet3D has fewer parameter counts, it is limited to 0.8 FPS due to its point cloud-based architecture. We evaluate two-stage frameworks where UniDet3D results are passed to VLMs for ordering, which achieves only 0.1 FPS and cannot meet the real-time demand. Meanwhile, although a standalone 2D detector such as YOLO can operate at 37.0 FPS, our robotic manipulation tasks require 3D oriented bounding boxes for effective performance. To establish a complete pipeline, we added a post-processing step that converts YOLO's

Table 8: Manipulation performance under **hard** setting in the simulation environment. Succ. Num., Fail Num., RC, OD and SR refer to number of successfully picked-up objects, number of failed pick-up attempts, residual count, object disturbance and success rate, respectively. Privilege indicates models are accessible to objects' ground truth information.

| Framework | Privilege | Succ. Num.↑ | Fail Num.↓ | RC↓ | OD↓ | SR(%)↑ |
|---|---|---|---|---|---|---|
| GPT-4o | ✓ | 43.2 | 17.7 | 16.8 | 12.0 | 71.4 |
| Claude-3.7-Sonnet | ✓ | 54.7 | 16.0 | 5.3 | 10.4 | 77.9 |
| Gemini-2.5 | ✓ | 54.2 | 17.4 | 5.8 | 12.4 | 78.5 |
| Gemma-3 | ✓ | 52.5 | 22.8 | 7.5 | 11.2 | 70.6 |
| InternVL3 | ✓ | 47.9 | 19.8 | 12.1 | 13.9 | 70.4 |
| Qwen2.5-VL | ✓ | 52.3 | 23.2 | 7.7 | 11.4 | 70.4 |
| UniDet3D+Confidence | ✗ | 50.3 | 64.8 | 9.7 | 8.4 | 46.1 |
| UniDet3D+Distance | ✗ | 45.5 | 141.9 | 14.5 | 12.8 | 24.9 |
| UniDet3D+GPT-4o | ✗ | 36.4 | 73.9 | 23.6 | 9.2 | 33.4 |
| UniDet3D+Claude-3.7-Sonnet | ✗ | 41.4 | 96.2 | 18.6 | 14.2 | 30.7 |
| UniDet3D+Gemini-2.5 | ✗ | 43.5 | 120.7 | 16.5 | 13.1 | 26.6 |
| UniDet3D+Gemma-3 | ✗ | 44.7 | 82.7 | 15.3 | 13.5 | 35.8 |
| UniDet3D+InternVL3 | ✗ | 43.2 | 62.8 | 16.8 | 11.4 | 43.1 |
| UniDet3D+Qwen2.5-VL | ✗ | 40.1 | 103.6 | 19.9 | 13.2 | 28.3 |
| OrderMind-Mini (Ours) | ✗ | 54.6 | 5.9 | 5.4 | 7.2 | 90.4 |
| OrderMind (Ours) | ✗ | 56.7 | 2.9 | 3.3 | 4.3 | 95.4 |

Table 9: Detection Result under easy, moderate and hard difficulty modes in simulation environment.

| Model | Modality | $mAP_{50}^{easy}$ ↑ | $mAP_{50}^{mod}$ ↑ | $mAP_{50}^{hard}$ ↑ |
|---|---|---|---|---|
| UniDet3D | Point Cloud | 46.3 | 36.7 | 33.6 |
| OrderMind (Ours) | RGB-D | 92.2 | 88.6 | 78.3 |

2D detections and 2D masks into 3D oriented bounding boxes. This involved extracting object point clouds from depth images according to the 2D predictions, followed by fitting 3D oriented bounding boxes using an optimized RANSAC algorithm. As a result, the full YOLO-based pipeline runs at 11.9 FPS. In comparison, our proposed OrderMind-mini model achieves a significantly higher speed of 21.3 FPS, making it a much more efficient solution for the actual manipulation task.

Tab. 10 compares our method against the same baselines. Across all difficulty levels, the proposed approach consistently achieves higher success rates (SR) and lower residual counts (RC), mirroring the trend observed in the main experiments. This consistency indicates that the VLM-generated sequential plans remain coherent and effective even in novel environments with unseen objects. A slight increase in Object Disturbance (OD) under the hard setting reflects minor contacts during grasping, yet tasks are still completed successfully without destabilizing the scene, this unlike the baseline methods, which often fail after such perturbations.

These results demonstrate that the proposed VLM-based label generation process generalizes well beyond the original setup, producing reliable and interpretable manipulation sequences across diverse scenes and object types.

# D   Additional Ablation Study

In this section, we present an ablation study to determine the optimal weight for the loss $\mathcal{L}order$, with results summarized in Table 11. We experimented with weights of 3, 5, 7, and 10. The data indicates that a weight of 5 achieves the best performance, yielding an RC of 1.0, an OD of 1.4, and an SR of 95.3%. Other weights resulted in comparatively inferior metrics. Thus, a weight of 5 was selected as the optimal value for $\mathcal{L}order$.

Table 10: Generalization performance of different frameworks across varied environments and object configurations. RC, OD and SR refer to residual count, object disturbance and success rate respectively. "Privileged" indicates methods with access to ground-truth object attributes.

| Framework | Privilege | Easy | | | Moderate | | | Hard | | | FPS |
|---|---|---|---|---|---|---|---|---|---|---|---|
| | | RC↓ | OD↓ | SR↑ | RC↓ | OD↓ | SR↑ | RC↓ | OD↓ | SR↑ | |
| GT+SPH | ✓ | 0.9 | 0.7 | 89.5 | 1.7 | 1.5 | 86.8 | 1.8 | 4.4 | 90.9 | / |
| GT + Gemini-2.5 | ✓ | 0.2 | 2.5 | 99.2 | 2.5 | 4.6 | 91.2 | 7.2 | 7.2 | 88.0 | 0.2 |
| YOLO11-seg+SPH | ✗ | 8.9 | 1.9 | 59.0 | 10.3 | 4.1 | 66.9 | 17.5 | 8.5 | 69.0 | 11.9 |
| YOLO11-seg+Gemini2.5 | ✗ | 9.2 | 1.9 | 60.8 | 13.8 | 4.4 | 65.4 | 21.6 | 8.7 | 67.5 | 11.9 |
| YOLO11-det+SPH | ✗ | 7.7 | 7.7 | 65.2 | 11.1 | 3.9 | 66.7 | 18.1 | 8.1 | 69.0 | 11.9 |
| YOLO11-det+Gemini2.5 | ✗ | 7.8 | 2.0 | 64.3 | 12.6 | 4.6 | 67.4 | 17.4 | 7.6 | 67.6 | 11.9 |
| Ours | ✗ | 4.2 | 1.8 | 85.3 | 5.3 | 4.0 | 83.2 | 7.4 | 11.2 | 83.7 | 21.3 |

Table 11: Ablation study on weight selection for loss $\mathcal{L}_{order}$. RC, OD and SR refer to residual count, object disturbance and success rate, respectively.

| Weight | RC ↓ | OD ↓ | SR(%)↑ |
|---|---|---|---|
| 3 | 3.7 | 3.9 | 89.6 |
| 5 | 1.0 | 1.4 | 95.3 |
| 7 | 2.6 | 3.5 | 91.3 |
| 10 | 2.8 | 3.6 | 92.2 |

# E   Order Label Generation Pipeline with Vision-Language Model

To mitigate potential inconsistencies in VLM-generated labels, we aggregate multiple rankings per scene using a Plackett–Luce model, which infers a stable global ordering from noisy pairwise comparisons. This ensemble step improves label reliability during training. We demonstrate the prompt given to VLM as follows:

Listing 1: Prompt for VLM

```
Image Prompt: An RGB image at a resolution of 1048*1024 with object
segment masks and ids, as demonstrated in Fig.7.

Text Prompt: Based on this image, imagine you are a robotic arm used
 to manipulate objects from a table. You need to manipulate objects
quickly and accurately while making sure the success rate is high.
The arm has a suction cup and its starting position is [1408, 1024],
 which is the bottom-right corner of the image. After manipulating
each object, you need to return to the starting position before
manipulating the next one. You must follow the rules in '
criteria_for_manipulation'. Criteria 1 has the highest priority, and
 the priorities of the others decrease accordingly. You can be
flexible as needed, but the ultimate goal is to quickly and
accurately collect all the objects. You must collect all the objects
! The object information template is in 'objects_template', where '
id' is the ID of each object. The output should be in JSON format as
 shown in 'output_format'. In 'ranking', write the final order,
where objects are sorted according to the manipulating order, from
first to last. All objects must be included and sorted correctly.
Think step by step, show the correct manipulating order. Strictly
follow my output format and instruction. Please analyze the input
data thoroughly and carefully. Ensure the ranking is derived from a
serious and deep evaluation of the available information.

"criteria_for_manipulation": {
    "criteria_1": "Manipulate objects whose <is_isolated> is true
    and <size> is large.",
    "criteria_2": "Manipulate objects whose <is_local_highest> is
    true and <relative_position_to_robot> is small.",
    "criteria_3": "Manipulate objects whose <rotation_degree> is
    small."
```

```
    },
"objects_template": {
    "id": "<id>",  # object id
    "is_isolated": "<is_isolated>",  # true / false
    "is_local_highest": "<is_local_highest>",  # true / false
    "size": "<size>",  # small / large
    "box_area": "<box_area>",  # small / large
    "rotation_degree": "<rotation_degree>", # small / large
    "relative_position_to_robot": "<relative_position_to_robot>",
    # float number
    },
"output_format": {
    "ranking": "<id> -> <id> -> <id> -> ... -> <id>"
    }.
```

## F  Details on Object Attributes

In this paper, object attributes are divided into intrinsic attributes and extrinsic attributes. Intrinsic attributes describe object's extent and category. Extrinsic attributes describe the object's pose in the world coordinate system, which includes the center point $p_w = [x, y, z]^T$ and the 3D orientation $\theta_w = [roll, pitch, yaw]^T$. First, we use different heads in the detector to predict the size information, the center point in the camera coordinate system $p_c$, and the 3D orientation in the object coordinate system $\theta_o$. Subsequently, we use the object-to-camera homogeneous transformation matrix $T_{oc} \in SE(3)$, and the camera-to-world homogeneous transformation matrix $T_{wc} \in SE(3)$ to convert center point and 3D orientation into the world coordinate system. The specific process is as follows:

$$\begin{bmatrix} p_w \\ 1 \end{bmatrix} = T_{wc} \begin{bmatrix} p_c \\ 1 \end{bmatrix}, \begin{bmatrix} \theta_w \\ 1 \end{bmatrix} = T_{wc} \cdot T_{co} \begin{bmatrix} \theta_o \\ 1 \end{bmatrix}.$$

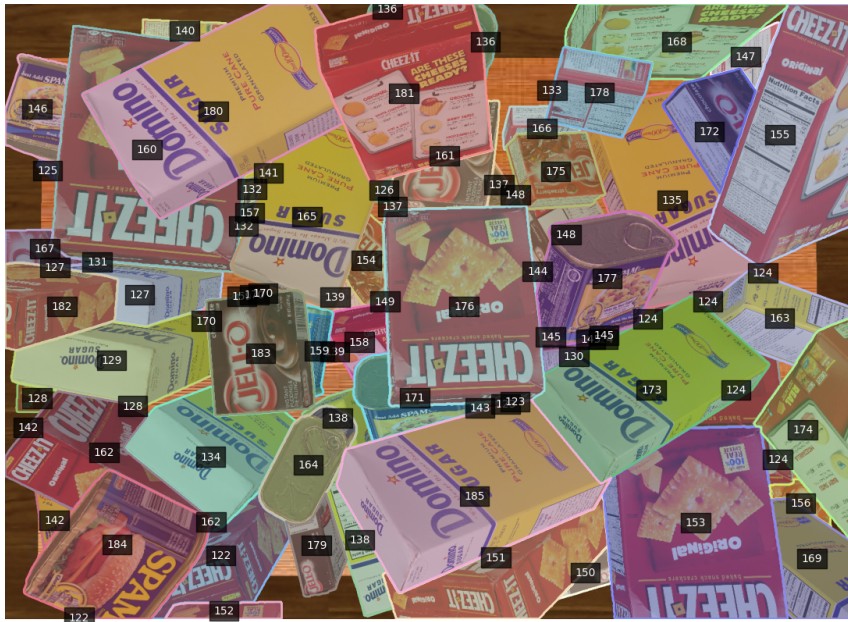

Figure 7: Example of an image prompt provided to the VLM, containing segment masks and object IDs marked on each object.

