# OpenReview forum: "Learning Spatial-Aware Manipulation Ordering"
_NeurIPS.cc/2025/Conference — NeurIPS 2025 poster_

### Official Review · Reviewer_RBro · 2025-06-21

**Clarity:** 3
**Significance:** 3
**Originality:** 3
**Rating:** 4
**Confidence:** 3

**Summary:**

In this paper, the authors address the problem of the order of manipulation, which is crucial in tasks such as picking up postal parcels. The robot must determine which item to handle first and which to handle next.

**Questions:**

1. The attributes are very important for representing the object. How are these attributes obtained? I think the authors should provide more details on this process.
2. What are the extrinsic attributes? What do the arrows with different colors represent, and how are they obtained?
3. In Fig. 2, do you directly combine all the embeddings together? How are the circle and rectangle combined? I believe there are several fusion methods; could the authors clarify which add is used here?
4. How is fusion implemented in the extrinsic module?

**Ethical Concerns:**

["NO or VERY MINOR ethics concerns only"]

**Final Justification:**

Thank you for your feedback. Although the method the authors have employed builds upon existing methods to tackle this issue, the technical novelty does not seem to be particularly high. Nonetheless, I believe the paper still carries some significance. However, the concerns raised by Reviewer 7B should be taken into consideration.

**Limitations:**

See weakness

**Quality:**

3

**Strengths And Weaknesses:**

The topic is useful, and the performance is good. The method considers both spatial and temporal contexts, which should be valuable for this problem. However, when examining the framework, it seems that the authors treat the problem primarily as a computer vision task, similar to how appreciation order is used to detect objects in an image. I wonder if this problem has been considered from the perspective of object detection. If Yes. Whether this method could be applied to the context in this paper to achieve even better results.

---

> ### Author Rebuttal · Authors · 2025-07-31
>
> We thank the reviewer for the detailed review as well as the suggestions for improvement. Our response to the reviewer’s comments is below:
>
> **W1: On framing the task as an object detection pipeline.**
>
> We sincerely thank the reviewer for this insightful question. We have implemented and evaluated two computer vision baselines built upon the YOLO11 model [1]. We use YOLO11-det as a detection baseline and YOLO11-seg as the segmentation baseline. To ensure a fair comparison, we adapted our Spatial Priors into a heuristic policy, which was described in Sec. 3.3 in our main paper, into a heuristic policy and name it as Spatial Priors Heuristic (SPH). As an effective heuristic policy, SPH can provide the detection model with a better sorting result.
>
> While a standalone 2D detector such as YOLO can operate at 37.0 FPS, our robotic manipulation tasks require 3D oriented bounding boxes for effective performance. To establish a complete pipeline, we added a post-processing step that converts YOLO’s 2D detections and 2D masks into 3D oriented bounding boxes. This involved extracting object point clouds from depth images according to the 2D predictions, followed by fitting 3D oriented bounding boxes using an optimized RANSAC algorithm. As a result, the full YOLO-based pipeline runs at 11.9 FPS. In comparison, our proposed OrderMind-mini model achieves a significantly higher speed of 21.3 FPS, making it a much more efficient solution for the actual manipulation task.
>
> **Table 1**. Success Rate (SR) of our model vs. simple heuristic baselines across difficulty modes. "Privileged" indicates methods with access to ground-truth object intrinsic and extrinsic attributes.
> | Method                                 | Privilege | SR-Easy ↑ | SR-Moderate ↑ | SR-Hard ↑ | FPS  |
> | -------------------------------------- | :-------: | :--------: | :------------: | :--------: | :--: |
> | GT + SPH        |     √     |    96.3    |      96.3      |    89.8    |  /   |
> | YOLO11-seg + SPH |     ×     |    75.1    |      77.5      |    76.4    | 11.9 |
> | YOLO11-det + SPH |     ×     |    75.5    |      79.8      |    74.9    | 11.9 |
> | **OrderMind (Ours)**           |     ×     |    94.2    |      89.6      |    90.4    | 21.3 |
>
> From the Table 1, our proposed model consistently outperforms the detection-based and segmentation-based baseline across all difficulty mode. The primary reason for this performance gap lies in the difference between a decoupled pipeline and our integrated end-to-end approach. The heuristic detect-then-sort method treats perception and ordering as two disconnected steps. The sorting heuristic operates only on the geometric outputs like bounding boxes, losing the rich visual and contextual information from the original image that is necessary for reasoning about complex spatial relationships.
>
> In contrast, our model learns to perceive objects and infer their manipulation order simultaneously. It does not simply detect objects but learns to understand the intricate spatial arrangements and dependencies between them, guided by the VLM's knowledge. This integrated approach allows our model to handle challenging scenarios that require long-horizon planning and spatial understanding, which are difficult for a simple heuristic to capture. By unifying object understanding and order prediction into a single inference pass, our model achieves more accurate and robust manipulation ordering, effectively maintaining spatial consistency across diverse and cluttered scenes.
>
> ---
>
> **Q1: Details on Attribute Acquisition.**
>
> In this paper, object attributes are divided into intrinsic attributes and extrinsic attributes. Intrinsic attributes describe object's extent and classification. Extrinsic attributes describe the object's pose in the world coordinate system, which includes the center point $p_{w}=[x,y,z]^T$ and the 3D orientation $\theta_{w}=[roll,pitch,yaw]^T$. First, we use different heads in the detector to predict the size information, the center point in the camera coordinate system $p_{c}$, and the 3D orientation in the object coordinate system $\theta_{o}$. Subsequently, we use the object-to-camera homogeneous transformation matrix $T_{oc} \in SE(3)$, and the camera-to-world homogeneous transformation matrix $T_{wc} \in SE(3)$ to convert center point and 3D orientation into the world coordinate system. The specific process is as follows:
> $$
> \begin{bmatrix} p_{\text{w}} \\\\ 1 \end{bmatrix} = T_{wc} \begin{bmatrix} p_{\text{c}} \\\\ 1 \end{bmatrix} , \begin{bmatrix} \theta_{\text{w}} \\\\ 1 \end{bmatrix} = T_{wc}\cdot T_{co} \begin{bmatrix} \theta_{\text{o}} \\\\ 1 \end{bmatrix}.
> $$
>
> ---
>
> **Q2: Details on extrinsic attributes.**
>
> We sincerely thank the reviewer for their meticulous observation. We deeply apologize for a labeling error in Figure 2. The labels for intrinsic attributes and extrinsic attributes at the right of the image backbone were inadvertently swapped and we will revise it in the final version.
>
> Extrinsic attributes refer to an object's 6D pose within the world coordinate system. This pose comprises the object's center point and its 3D orientation. These attributes vary as the object's position changes within the world coordinate system.
> Arrows of different colors represent the object's orientation angles in the world frame. The RGB colors correspond to the x, y, and z axes, respectively. To determine the 3D orientation, our detector head first predicts the object's 3D orientation $\theta_{o}=[roll,pitch,yaw]^T$, relative to the camera coordinate system. Next, we use the object-to-camera homogeneous transformation matrix $T_{oc} \in SE(3)$, and the camera-to-world homogeneous transformation matrix $T_{wc} \in SE(3)$ to convert 3D orientation into the world coordinate system as the following equation:
> $$
> \begin{bmatrix} \theta_{\text{w}} \\\\ 1 \end{bmatrix} = T_{wc}\cdot T_{co} \begin{bmatrix} \theta_{\text{o}} \\\\ 1 \end{bmatrix}.
> $$
>
> ---
>
> **Q3: Clarification on the embedding fusion method in Figure 2.**
>
> We would like to clarify that we combine all the embeddings using element-wise addition. The embeddings represented by rectangles in Figure 2 correspond to the intrinsic and extrinsic embeddings of the objects. The circles refer to the object order queries. All these embeddings are projected into the same feature dimension before being combined. More specifically, the intrinsic embedding is calculated by encoding the 3D extent of the object using a sinusoidal positional encoding, while the extrinsic embedding is derived from the robot-object's relative position using the same encoding method. Both are mapped to the same feature dimension as the object order query. For the visual embedding, the object RoI queries from the image backbone are passed through an MLP to match the object order query's feature dimension.
>
> Regarding the fusion methods, we have tried a few simple alternatives, but we observed that the final results were similar. For reasons of simplicity and computational efficiency, we ultimately chose element-wise addition as our fusion strategy. In the future, we will explore more advanced or complicated fusion approaches to see if they further improve performance.
>
> ---
>
> **Q4: Details on the implementation of fusion in the extrinsic module.**
>
> We use the spatial graph from object-centric point clouds as introduced in Section 3.3 in our main paper. The fusion process consists of two main steps. First, for each object represented by its center point $p_i$ and feature $f_i$, we identify its spatial neighbors using a k-Nearest Neighbors strategy. This allows us to capture the local geometric relationships and context for every object. Next, for each object, we perform feature fusion by concatenating the feature $f_i$ with the difference $f_j - f_i$ from each neighboring object $p_j$ in the k-nearest set $N_k(i)$. This concatenated feature is then processed by a linear layer, and finally, we aggregate over all neighbors using a max pooling operator $M$, as described in as follows:
> $$
> Fusion(f_i) = M(Linear(Concate(f_i, f_j-f_i)),\forall p_j \in N_k(i).
> $$
>
> This fusion mechanism is motivated by the need to encode the physical interaction and alignment of objects, which can represent how objects affect each other's manipulability in the scene. By constructing these spatially-aware object embeddings, our approach improves the model's ability to perceive the local scene structure. These fused features are subsequently combined with visual representations to better support downstream ranking tasks.
>
> **Reference**
>
> [1] Glenn Jocher and Jing Qiu. Ultralytics YOLO11. 2024

---

### Official Review · Reviewer_84U8 · 2025-06-21

**Clarity:** 3
**Significance:** 3
**Originality:** 3
**Rating:** 5
**Confidence:** 3

**Summary:**

This paper introduces OrderMind, a spatial-aware framework for learning object manipulation order in cluttered scenes. The idea is to directly learn manipulation priority from RGB-D inputs using a combination of a k-NN spatial graph (for object-object and object-robot relationships) and a temporal ordering module that ranks objects by learned priority scores. Labels for supervision are generated using a vision-language model (VLM), but enhanced with spatial priors (like object isolation or stackability). The authors also built a large-scale benchmark (160k+ samples) and show both strong sim and real-world performance compared to heuristics, VLM pipelines, and two-stage baselines.

**Questions:**

Can you share examples of the prompts used for VLM ordering supervision? Was any spatial descriptor, e.g., relative pose or bounding box info, passed into the prompt?
The VLM-generated labels are critical to training. How consistent are these labels across different scenes and object types? Have you tested generalization to new objects or unseen environments beyond YCB?
What kind of errors does the model tend to make in hard real-world scenes (e.g., with deformable or occluded bags)? Qualitative failure examples would be valuable.

**Ethical Concerns:**

["NO or VERY MINOR ethics concerns only"]

**Final Justification:**

The authors' rebuttal comprehensively addresses my concerns with concrete details: full VLM prompt examples, new experiments validating label consistency/generalization in novel environments/objects, noise robustness ablation, and failure mode quantification from real video. Minor unresolved: Deeper analysis of VLM biases in rare edge cases (e.g., human validation subset) and expanded real-world testing on deformables/occlusions could further improve, but these are low-weight given the core strengths.Overall, the rebuttal elevates quality/reproducibility, outweighing remaining issues.

**Limitations:**

yes

**Paper Formatting Concerns:**

at line 464: We have state the challenge -> We have stated the challenge

**Quality:**

2

**Strengths And Weaknesses:**

**Strengths**: The problem that author explores is real and under-explored by other works. OrderMind speedup on slow multi-stage pipelines by jointly encoding spatial reasoning and prioritization in one model. Use of k-NN spatial graphs and attention-based temporal priority modeling is a well-motivated and effective architectural choice. No manual annotation needed. Proposed a new benchmark. Both qualitative and quantitative results show consistent improvements across difficulty levels and across simulation/real-world settings.

**Weaknesses**: Spatial priors help, but the generated order labels may still reflect unintended biases or fail in edge cases. Some analysis on this would help.
The authors miss details on VLM prompting: What exactly is the prompt format for order generation? Is there chain-of-thought prompting or just spatial descriptor input? Since the VLM is a core component for supervision, this should be shown.
In addition, the author should also discuss the label noise come from VLM. What happens when VLM labels are wrong or inconsistent? The model likely learns around it, but it'd be good to quantify how robust it is.

---

> ### Author Rebuttal · Authors · 2025-07-31
>
> We thank the reviewer for the detailed review as well as the suggestions for improvement. Our response to the reviewer’s comments is below:
>
> **W1 Q1: Prompt format and info for VLM ordering supervision.**
>
> Our prompt format has two components: a pre-processed image prompt and a meticulously designed text prompt. This format integrates both spatial descriptors and chain-of-thought (CoT) prompting.
>
> The image prompt is an RGB image at a resolution of 1024×1408. We annotate each object in the image with its segment mask and ID. In cluttered scenes, VLMs struggle to interpret spatial and stacking relationships from the raw image. To solve this, we overlay the segment masks and object IDs directly onto the image. This allows the VLMs to associate visual objects with textual descriptors and better capture relevant spatial information.
>
> Our text prompt integrates a task description, object-specific information, and grasping rules, utilizing both spatial descriptors and CoT prompting. First, we provide a textual description of the task scene and the robot's grasping process to help the VLM comprehend the goal. Instead of directly inputting large amounts of numerical data, we convert object information into spatial descriptors, making the spatial context easier for the VLM to understand.
>
> Based on this object information, we establish grasping rules with priorities. Since scenes with many objects can satisfy multiple rules simultaneously, we introduce CoT prompting. This guides the VLM to flexibly combine the rules with their priorities and to think step-by-step. To accelerate the speed of label generation, we do not require the VLM to output its detailed inference process.
>
> Due to the space limits, we will show our prompt example in the discussion stage.
>
> ---
>
> **Q2：Consistency of VLM-generated labels across scenes and objects.**
>
> We sincerely thank the reviewer for this insightful question. The consistency of VLM-generated labels is indeed critical to the generalizability of our method and we agree it is an important aspect to validate.
>
> We conducted a new set of experiments in a challenging environment. First, we designed a new simulation environment that significantly departs from the one used in our paper. We replaced the flat tabletop with *a deep-sided metal tray*. This change introduces not only new visual textures and backgrounds but also different physical constraints for manipulation. Second, we used a completely new set of five objects with diverse shapes and categories including *a laundry bottle with an integrated handle*, *a thin correction fluid pen*, *a cube-shaped tea box*, *a cylindrical hair conditioner bottle*, and *a rectangular book*. These objects are distinct from the relatively uniform box-like objects used in our original experiments. To further increase the complexity, we applied random scaling between 0.8 and 1.2 to the size of these new objects. The table below compares our method against the same baselines.
>
> **Table 1.** Comparison of different frameworks. RC, OD and SR refer to residual count, object disturbance and success rate respectively. "Privileged" indicates methods with access to ground-truth object intrinsic and extrinsic attributes.
> |Framework| Privilege|Easy RC ↓|Easy OD ↓|Easy SR ↑|Moderate RC ↓|Moderate OD ↓|Moderate SR ↑|Hard RC ↓|Hard OD ↓|Hard SR ↑|FPS|
> | :-------------------- | :-------: | :-------: | :-------: | :-------: | :-----------: | :-----------: | :-----------: | :-------: | :-------: | :-------: | :--: |
> |GT+SPH|√|0.9|0.7|89.5|1.7|1.5|86.8|1.8|4.4|90.9|/|
> |GT+Gemini-2.5|√|0.2|2.5|99.2|2.5|4.6|91.2|7.2|7.2|88.0|0.2|
> |YOLO11-seg+SPH|×|8.9|1.9|59.0|10.3|4.1|66.9|17.5|8.5|69.0|11.9|
> |YOLO11-seg+Gemini-2.5|×|9.2|1.9|60.8|13.8|4.4|65.4|21.6|8.7|67.5|11.9|
> |YOLO11-det+SPH|×|7.7|7.7|65.2|11.1|3.9|66.7|18.1|8.1|69.0|11.9|
> |YOLO11-det+Gemini-2.5|×|7.8|2.0|64.3|12.6|4.6|67.4|17.4|7.6|67.6|11.9|
> |**Ours**|×|4.2|1.8|85.3|5.3|4.0|83.2|7.4|11.2|83.7|21.3|
>
> The results show that our method consistently achieves a higher success rate and a lower residual count  across all difficulty levels. This performance trend aligns with our findings in Table 1 of the paper. This consistency suggests that the VLM was able to generate high-quality and coherent sequential plans for this novel and more complex scenario. Notably, our method exhibits a slightly higher Object Disturbance in the hard mode. However, this is coupled with a much lower RC and a higher SR. This indicates that while our policy may cause minor contact with cluttered objects during a grasp, it successfully completes the task without knocking them out of the robot workspace. In contrast, the baseline methods often fail entirely after such disturbances. This behavior highlights the robustness of the policy learned from the VLM-generated labels.
>
> In summary, these new experiments provide evidence that our VLM-based label generation process is not confined to a specific setup. It consistently produces effective and logical sequential labels for varied scenes and object types, confirming the generalizability and robustness of our approach.
>
> ---
>
> **W2: Robustness to label noise from VLMs.**
>
> We thank the reviewer for this excellent question regarding the potential for label noise from VLMs and our model's robustness.
>
> In our experiments in the main paper, we designed our label generation process specifically to address the possibility of noisy or inconsistent VLM outputs to some extent. Specifically, our method first generates multiple rankings for one scene by many turns of VLM labeling. We then ensemble these rankings using a Plackett-Luce model, which allows us to infer a more stable and reliable global ranking from various potentially noisy inputs. This step is our primary mechanism for improving the quality of the training labels and mitigating the impact of occasional VLM errors.
>
> While this ensemble approach helps to reduce noise at the source, we believe that a direct quantification of the model's robustness is a valuable addition. We started with our refined VLM-generated labels and then intentionally introduced controlled levels of noise by randomly swapping a certain percentage of pairwise orders. We ran this experiment on the 15,000 samples from the new scene mentioned in our response to **Q2**. The results are presented in the table below.
>
> **Table 2**. Ablation on model robustness to label noise. "Privileged" indicates methods with access to ground-truth object  intrinsic and extrinsic attributes.
> | Noise  Ratio | Privilege | SR-Easy↑ | SR-Mod↑ | SR-Hard↑ |
> | ------------ | :---------: | :--------: | :-------: | :--------: |
> | 0  | × | 85.35 | 83.19 | 83.66|
> | 10%| ×| 85.10 | 82.71 | 79.13|
> | 20%| × | 80.08  | 79.00 | 76.30 |
> | 50%| × | 78.88  | 76.91 | 73.81 |
> | 70%| × | 75.99 | 70.36 | 67.31 |
> | GT + random| √  | 75.50| 75.58| 68.88|
>
> GT+random denotes the success rate when using a random manipulation policy given the ground-truth intrinsic and extrinsic attributes of the object. As the table indicates, the performance degrades slightly with moderate noise. Under the extreme 70% noise condition, the performance degrades more substantially. We hypothesize that this occurs because, at sufficiently high noise ratios, our model is unable to extract meaningful information and thus essentially degenerates into a random policy. In scenarios of moderate difficulty, the success rate of our model falls due to errors introduced by perception noise. Thanks to both our label ensemble strategy and the unified learning of perception and ordering, our method is robust to VLM label noise.
>
> ---
>
> **Q3: Generalization to new objects and unseen environments.**
>
> Yes, we have tested the generalization ability of our method to new objects and unseen environments beyond the YCB dataset.
>
> The experiments presented in our main paper were specifically designed to evaluate performance in unseen environments. To clarify our experimental setup, the model was trained exclusively on data collected from a single dynamic conveyor belt in a factory. We then deployed and tested this model in two different and unseen settings. The first was a static scene within our laboratory for demonstrating, and the second was a different dynamic conveyor belt with various types of objects in the factory. These test environments featured notable variations from the training scene, including different lighting conditions, backgrounds, and degrees of motion blur. We believe this setup provides a direct evaluation of our model's ability to generalize to new environments and new objects.
>
> Furthermore, as described in our response to **Q2**, we set up a completely new scenario with a collection of novel objects that have diverse shapes. Our method continued to achieve a high success rate in manipulation tasks, and the object residual count and disturbance metrics remained low. These results are consistent with the results in our main paper.
>
> ---
>
> **Q4：Failure case in real-world scenes.**
>
> We manually analyzed a 30-minute video of our robotic arm's operation to categorize and quantify failures. Inaccurate object perception, challenges with deformable objects, and incorrect manipulation ordering emerged as the main issues.
>
> Specifically, regarding perception errors, inaccurate 3D rotation prediction contributed to 21% of the failures, while incorrect object center point identification accounted for another 15%. For challenges with deformable objects, an inability to find suitable suction surfaces caused another 21%. Furthermore, incorrect manipulation ordering led to object disturbances, accounting for 39%. The remaining 4% were attributed to other factors, such as poor synergy between the robotic arm and the camera. For instance, the robotic arm occasionally blocked the camera's view, which adversely affected the perception process. In our revised version, we will incorporate qualitative failure cases and we will focus on resolving these issues in our future work.

---

> ### Author Response · Authors · 2025-08-01
> **Supplementary: Prompt Example**
>
> As mentioned in our rebuttal, we are providing the prompt example here for reference due to previous space limitations:
>
>
>
> ```
> [Image Prompt]: An RGB image at a resolution of 1048*1024 with object segment masks and ids.
>
> [Text Prompt]: Based on this image, imagine you are a robotic arm used to manipulate objects from a table. You need to manipulate objects quickly and accurately while making sure the success rate is high. The arm has a suction cup and its starting position is [1408, 1024], which is the bottom-right corner of the image. After manipulating each object, you need to return to the starting position before manipulating the next one. You must follow the rules in 'criteria_for_manipulation'. Criteria 1 has the highest priority, and the priorities of the others decrease accordingly. You can be flexible as needed, but the ultimate goal is to quickly and accurately collect all the objects. You must collect all the objects! The object information template is in 'objects_template', where 'id' is the ID of each object. The output should be in JSON format as shown in 'output_format'. In 'ranking', write the final order, where objects are sorted according to the manipulating order, from first to last. All objects must be included and sorted correctly. Think step by step, show the correct manipulating order. Strictly follow my output format and instruction. Please analyze the input data thoroughly and carefully. Ensure the ranking is derived from a serious and deep evaluation of the available information.
>
> "criteria_for_manipulation": {
>      "criteria_1": "Manipulate objects whose <is_isolated> is true and <size> is large.",
>      "criteria_2": "Manipulate objects whose <is_local_highest> is true and <relative_position_to_robot> is small.",
>      "criteria_3": "Manipulate objects whose <rotation_degree> is small."
>     },
>
> "objects_template": {
>      "id": "<id>",  # object id
>      "is_isolated": "<is_isolated>",  # true / false
>      "is_local_highest": "<is_local_highest>",  # true / false
>      "size": "<size>",  # small / large
>      "box_area": "<box_area>",  # small / large
>      "rotation_degree": "<rotation_degree>", # small / large
>      "relative_position_to_robot": "<relative_position_to_robot>",  # float number
>     },
>
> “output_format”: {
>      "ranking": "<id> -> <id> -> <id> -> ... -> <id>"
>     }.
>
> ```
>
> Our prompt includes an overlaid image prompt with segment masks and object IDs, and a text prompt that describes the task, objects information and manipulation criteria with priorities. We use chain-of-thought prompting to guide the VLMs in step-by-step reasoning.

---

> > ### Comment · Reviewer_84U8 · 2025-08-04
> >
> > Thank you to the authors for the thorough rebuttal and supplementary materials. The prompt examples clarify the use of overlaid masks, spatial descriptors, CoT, and prioritization rules, which address my concern. I hope authors can also make sure include it in the final version. The new experiments on label consistency in varied environments/objects, the noise robustness ablation, the clarification of generalization beyond YCB are helpful for me the analysis this work better. These additions address my concerns effectively, particularly around VLM supervision transparency, consistency, and model robustness. While some edge-case VLM biases could still be explored (e.g., via targeted human checks), and real-world testing expanded to more deformables/occlusions, the work now feels more complete. I have modified my score accordingly.

---

> > > ### Author Response · Authors · 2025-08-05
> > >
> > > Thank you very much for your positive and constructive feedback. We appreciate your suggestions and will make sure to incorporate the necessary clarifications and additions in the final version. Your comments are very valuable to further improve our work.

---

### Official Review · Reviewer_7BAs · 2025-07-02

**Clarity:** 2
**Significance:** 2
**Originality:** 2
**Rating:** 3
**Confidence:** 3

**Summary:**

The agent should be able to look at a scene, understand 3D relationships between objects, and then take a series of manipulation actions to pick up unobstructed objects efficiently.

**Questions:**

1. I apologize to the authors if I've missed something critical but, its easy and fast to predict bounding boxes and masks, so it should be easy to simply rank unoccluded objects and then sort by proximity. Particularly because we are only working with overhead views.  Why is robot pose necessary here or even a bespoke architecture?

2. The fact that in hard scenarios the results only drop to 76% seems to validate my concern that the task is not that difficult to code up? Is this due to the myriad number of ways to succeed? Is there really a good reason to choose one ordering over another?

3. The real world examples appear incredibly simple.

**Ethical Concerns:**

["NO or VERY MINOR ethics concerns only"]

**Final Justification:**

The results presented in the rebuttal are crucial to the paper's claims and yet, still incomplete.  I recognize that more comparisons, particularly ones with visuals, cannot be included in the rebuttal but they should have been included in the original manuscript. As I noted, I can come up with a bunch of other simpler baselines, or reasons these particular comparisons don't match their system, but the paper should identify these and justify. each reply begins to give a little of what's necessary, but I still think the authors devised an approach and moved forward with publication without thinking deeply about the space or implications and are only now starting to perform analysis of useful baselines.

**Limitations:**

See questions -- more generally I'm unclear on what the constraints are on the system, what the failures look like, what would be necessary to address them, etc.

One angle I could see is that there's an efficiency concern which limits the use of large VLMs but then I would still expect to see a few simple mask/bbox/yolo/... style baselines that can run at very high frequencies.

**Quality:**

3

**Strengths And Weaknesses:**

The application domain is straight forward to understand and the task is of real utility. There are a number of points where I'm confused by the experimental setup and goals.

What I understand, is that a 2D image is mapped to 3D with biases and preferences like picking up two objects near each other is prioritized and the evaluation is done over varying amounts of clutter until models begin to fail.
What I don't understand is why a heuristic policy (or simple pipeline) would ever fail.  Additional motivation, analysis, and insights would be greatly appreciated.

---

> ### Author Rebuttal · Authors · 2025-07-31
>
> We thank the reviewer for the detailed review as well as the suggestions for improvement. Our response to the reviewer’s comments is below:
>
> **W1: Analysis on heuristic policy's failure case.**
>
> We sincerely thank the reviewer for raising this excellent question. It addresses the core motivation for our work, and we appreciate the opportunity to elaborate on why heuristic policies falter in complex, realistic scenarios.
>
> The main weakness of simple heuristics, such as always choosing an unoccluded object, is that they ignore complex physical interactions like stability and support. These approaches treat objects as independent and overlook crucial consequences of actions. For example, in a scenario with stacked objects, a heuristic might pick a tall, unoccluded object, but if that object supports others, removing it could destabilize the whole pile and cause a collapse. This shows how a locally optimal choice from a simple rule can have disastrous global effects.
>
> Our method aims to address this challenge by distilling the commonsense physical reasoning of a large Vision-Language Model (VLM) into a compact and efficient policy [1]. We employ the VLM as an expert teacher to generate spatial-aware optimal manipulation action labels for a given scene. A smaller, faster model is then trained on these labels to approximate the VLM's underlying logic about physical stability and object interdependencies, rather than memorizing simple geometric heuristics. This enables our manipulation policy to anticipate and prevent potential issues like the cascading collapses discussed earlier, making it more robust. We believe this distillation of reasoning is the key to overcoming the fundamental limitations of heuristic approaches, and we will clarify this motivation in the revision.
>
> ---
>
> **Q1: On the necessity of our model over a simple pipeline.**
>
> We thank the reviewer for this sharp and practical question. A simple pipeline approach is indeed an intuitive baseline, and we appreciate this opportunity to clarify why a more integrated model like ours is necessary for achieving robust and efficient manipulation.
>
> First, addressing the detection component, while predicting 2D boxes is fast, robotic grasping fundamentally requires precise 3D orientation. As demonstrated in **Tab. 1 in response to W1 for reviewer RBro**, a pipeline that starts with 2D detections would need an additional step to estimate the 3D pose from 2D masks and depth data. Our model streamlines this by directly and efficiently predicting the necessary 3D information in one go. Second, a heuristic policy that simply sorts unoccluded objects by proximity is insufficient for two main reasons. As we demonstrated in our response to the **Q2 for reviewer Erw7**, it ignores complex physical stability issues, such a pipeline is inherently short-sighted. As demonstrated in the **Tab.1 for reviewer Erw7**, if model inference is run only once for every five manipulation attempts, our model’s performance remains high because it has learned to plan ahead. The heuristic pipeline’s success rate, however, collapses, as it lacks any foresight beyond the immediate next step.
>
> The robot's pose is included to help the model consider motion-related risks beyond simple proximity. Using the robot's fixed XYZ base, we calculate its position relative to each object, allowing the model to balance motion cost with physical outcomes. For instance, the model can prefer a slightly farther, stable object over a closer, unstable one if it reduces the risk of destabilizing the pile.
>
> In summary, we do not simply model the manipulation ordering problem as a decomposable task of "detect-then-sort". It demands jointly reasoning about object geometry, physical stability, and robot-object mutual constraints.
>
> ---
>
> **Q2: Regarding task difficulty and the significance of ordering with multiple solutions.**
>
> Regarding the 76% success rate in hard scenarios: In our experiments, the Easy scenario features relatively simple stacking and minimal overlap between objects, resulting in a higher success rate. In the Moderate scenario, increased complexity in stacking and occlusion causes the success rate to drop to 78.5%. Although the number of objects further increases in the Hard scenario, its stacking relationships are similar to those in the Moderate scenario, with the main challenge still coming from complex stacking. Thanks to the model's real-time inference capability, we can effectively handle various disturbances in the scene. As a result, even with a much larger number of objects, the success rate only slightly drops from 78.5% to 76.6%, demonstrating the model's robustness and adaptability to task complexity. This high success rate is not due to the simplicity of the task itself, but rather to the model’s strong ability to handle such complex scenarios.
>
> Regarding the selection of grasping order: While there may not always be one optimal order, the multiple successful sequences possible actually highlight our model's flexibility. Even when several objects share high priority, our model finds rational grasping orders. As shown in Table 1 in the main paper, our method outperforms heuristic and VLM-based policies by maximizing success rates, minimizing disruption, and efficiently reducing remaining objects. Thus, even without a "best" order, the model ensures reliable, efficient operation.
>
> ---
>
> **Q3: The real world examples appear incredibly simple.**
>
> The scenes in Figure 5 in the main paper are indeed from a controlled laboratory setting, which we created to clearly and intuitively demonstrate our model's core effectiveness. We have successfully deployed and validated our model in a real-world factory environment, where the conditions are substantially more cluttered and complex than even the 'hard' scenarios shown in Figure 4. To better showcase our method's performance in these practical conditions, we will replace Figure 5 in the final version of the paper with more complex, anonymized images from this real factory deployment. Furthermore, we will update our supplementary video to include anonymized footage from these challenging, real-world factory environments.
>
> ---
>
> **Q4: The constraints and the failures of our system.**
>
> Our system faces key constraints: limited computational resources (15 TFLOPS on the edge device), a 30 ms time requirement, and strict safety rules— all inference must be done locally without cloud processing. Reviewing 30 minutes of robotic arm operation, we found four main failure sources. Perception errors were primary contributor, with 21% from inaccurate 3D rotation and another 15% from object center point misidentification. For challenges with deformable objects, an inability to find suitable suction surfaces caused 21% of the failures. Furthermore, incorrect manipulation ordering led to object disturbances, accounting for 39%. Other factors, like poor synergy between the robotic arm and the camera, accounted for 4%.
>
> We intend to replace our current CNN-based backbone with a transformer-based one to enhance the model's understanding of object pose. We will also explore reinforcement learning and similar methods to optimize robotic arm motion paths, thereby reducing disturbances to other objects. Furthermore, we will refine the interaction logic between the camera and the robotic arm to improve their overall synergy.
>
> ---
>
> **Q5: Comparison with fast heuristic baselines.**
>
> We sincerely thank the reviewer for this constructive suggestion. We have implemented and evaluated two high-speed baselines built upon YOLO11 model, one using its detection outputs (YOLO11-det) and another using its segmentation masks (YOLO11-seg), as demonstrated in the **Tab. 1 in response to W1 for reviewer RBro**.
>
> First, a standalone 2D detector like YOLO can achieve 37.0 FPS on its own, However, our robotic manipulation task fundamentally requires 3D oriented bounding boxes to execute manipulation. Therefore, a complete baseline pipeline must include a subsequent step to convert these 2D outputs into 3D information. For this, we extracted object point clouds from the depth image using the 2D predictions and then fitted 3D oriented boxes with a highly optimized RANSAC algorithm. The final end-to-end speed of this YOLO-based pipeline is 11.9 FPS. This is slower than our proposed OrderMind-mini, which operates at 21.3 FPS. For the actual task at hand, our model provides a much faster solution.
>
> Second, our method also demonstrates substantially better task performance. The YOLO-based baselines exhibit a lower manipulation success rate and leave a larger number of objects in the scene. This performance gap stems from two fundamental issues. The first is the error propagation in a sequential pipeline. The 2D-to-3D fitting process inevitably introduces inaccuracies in the resulting 3D bounding boxes, leading to grasp failures. The second and more critical issue is that such a baseline lacks a deep understanding of spatial relationships. It can only rely on simple heuristics for ordering, which is insufficient for resolving complex occlusions and dependencies.
>
> This comparison highlights the advantage of our work. Instead of approaching the problem as a simple detection-then-sorting pipeline, our method leverages VLM-guided supervision to learn spatial awareness and ordering in an end-to-end fashion. Our model does not just detect objects. It comprehends the relationships between them to infer a valid and efficient manipulation order. This capability proves crucial for succeeding in long-horizon and orderly manipulation tasks where simpler heuristic-based approaches falter.
>
> **Reference**
>
> [1] Zhou W, Tao M, Zhao C, et al. Physvlm: Enabling visual language models to understand robotic physical reachability. CVPR2025

---

> > ### Comment · Reviewer_7BAs · 2025-08-06
> > **Thank you for expanding**
> >
> > Thank you for expanding your experiments and comparisons. These dramatically strengthen the work.  However, the performance of the weakest models are still very strong which worries me about the data/task, and the inclusion of additional comparisons doesn't meet the bar we should expect in terms of identifying facets of the problem domain, clarifying what is required to reason about each, and why the baseline/heuristic is designed for this yet insufficient.  I can't judge if the LM is failing to understand the predictions vs a heuristic of IOU between mask and BBox, or I could propose that maybe a simple height based ordering after depth estimation would be the best baseline, etc.  The paper should identify these possibilities and evaluate or refute them.

---

> > > ### Author Response · Authors · 2025-08-06
> > >
> > > We thank the reviewer for the insightful follow-up. To clarify the purpose and contributions of our work, we address the main points as follows:
> > >
> > > **1. Significant Performance Improvements**
> > >
> > > As shown in *Tab. 1 in response to reviewer RBro*, the weakest baseline model already achieves a 74.9% success rate on the hard mode, which is considered strong in academic settings. However, our distilled model improves this result substantially, reaching a 90.4% success rate under identical conditions. Similarly, in *Tab. 1 for reviewer 84U8*, we construct a completely novel scenario where the baseline achieves 67.5%. Again, our model demonstrates consistent superiority, attaining 83.7% success. These results **validate the effectiveness of our approach across multiple settings and scenarios**.
> > >
> > > **2. On the Spatial Priors Heuristic (SPH): Purpose and Height-based Baseline**
> > >
> > > We appreciate the reviewer’s suggestion to compare against a simple height-based ordering policy as a baseline. The height-based policy, which prioritizes the object with the highest Z coordinate, performs reasonably well for vertically stacked objects but **fails in more commonly-seen *tilted support* scenarios**. For example, as shown in the following figure, the policy may incorrectly remove supporting object B before supported object A, simply because B’s center of mass or highest point is above A. This often causes the structure to collapse.
> > >
> > > ```
> > >              _ _ _ _ _ _ _ _ _ _ _ _ _
> > >            /      / \   _ _ _   table/
> > >           /  / \ /   \ |     |      /|
> > >          /  /   \ B   \|    _|_    /||
> > >         /  / A  / *  /_|_ _|_  |  / ||
> > >        /  / *  /    /| other |_| /
> > >       /  /    /\   / |objects|  /
> > >      /   \   /  \ /__|_ _ _ _| /
> > >     /      \/     |__|        /
> > >    /_ _ _ _ _ _ _ _ _ _ _ _ _/
> > >    ||                       ||
> > > ```
> > >
> > > In our work, the spatial priors were originally designed to serve as a prompt policy for the VLM during order label generation, as depicted in Sec. 3.3 in our main paper, rather than as a heuristic for deployment. For the purpose of ablation and comparison, we also evaluated spatial priors as a heuristic policy named SPH. SPH combines two strategies:
> > >
> > > * First, it identifies and removes isolated objects,
> > > * Then, it applies height-based ordering to the remaining objects.
> > >
> > > **The SPH encompasses the height-based baseline**, as it first reduces complexity by clearing easily accessible objects. As shown in the following table, **SPH outperforms the naive height-based policy**, by clearing isolated objects first, SPH creates more space to make the subsequent manipulation of complex, interdependent structures safer and more likely to succeed. It accepts minor, non-catastrophic disturbances in service of clearing more objects, thus achieving a higher overall success rate and a lower final residual count. Nonetheless, both SPH and the simple height-based policy share the same limitation when dealing with tilted support scenarios.
> > >
> > > **Table. 1** Comparison of height-based policy and our SPH policy. "Privileged" indicates methods with access to ground-truth object intrinsic and extrinsic attributes.
> > >
> > > | Framework   | Privilege | Residual Count (Mod )↓ | Object Disturbance(Mod)↓ | Success Rate(Mod)↑ | Residual Count(Hard)↓ | Object Disturbance(Hard)↓ | Success Rate(Hard)↑ |
> > > | ----------- | :-------- | ---------------------- | ------------------------ | ------------------ | --------------------- | ------------------------- | ------------------- |
> > > | GT + height | √| 2.0| 2.5| 85.2| 4.4| 3.7| 82.1|
> > > | GT + SPH    | √| 1.0| 2.6| 96.3| 2.5| 4.3| 89.8|
> > >
> > > **3. From Heuristic Policies to VLM-based Reasoning**
> > >
> > > As SPH cannot reason qualitatively about nuanced physical stability. At this point, one could try to engineer ever more sophisticated heuristic rules. Instead, we **view the VLM-based approach as a fundamentally different and more scalable alternative**. Our approach leverages VLMs’ emergent commonsense physical reasoning to infer safe manipulation orders. VLMs can, in principle, overcome many limitations of fixed geometric heuristics. However, direct use of VLMs is impractical in deployment due to their slow inference, high computational cost, and issues such as multi-object hallucination [1] and visual grounding errors [2].
> > >
> > > In our work, we distill the powerful but slow and resource-intensive reasoning abilities of VLMs into a compact and efficient model. By training on ensembled labels produced via VLMs guided with SPH-augmented prompts, our learned model acquires spatial awareness and ordering strategies in an end-to-end manner. This distillation approach allows us to achieve a 90.4% success rate in hard scenarios at 21.3 FPS, as is demonstrated in the *Tab. 1 in response to reviewer RBro*.
> > >
> > >
> > > [1] Chen X, Ma Z, Zhang X, et al. Multi-object hallucination in vision language models. NIPS 2024
> > >
> > > [2] Zeng Y, Huang Y, Zhang J, et al. Investigating compositional challenges in vision-language models for visual grounding. CVPR 2024

---

### Official Review · Reviewer_Erw7 · 2025-07-03

**Clarity:** 3
**Significance:** 3
**Originality:** 3
**Rating:** 5
**Confidence:** 4

**Summary:**

The authors propose a neural network-based approach for the manipulation task of ordered object picking. The use an image backbone on RGB-D images to predict object bounding boxes together with attributes. On top of the bounding boxes and attributes, a neural network is trained to predict picking priority score, which is then used to create an execution plan. The approach is validated both in simulation and real-world experiments, and shows impressive performance in picking success rate, significantly outperforming VLM baselines.

**Questions:**

- What is the architecture of the image backbone? I could not find details neither in the paper nor in supplementary.
- does OrderMind create a single plan and execute it in simulation and real-world experiments, or replan every few frames? in case of the latter, are plans stable for rearrangement? is there index shuffle during replan?

**Ethical Concerns:**

["NO or VERY MINOR ethics concerns only"]

**Final Justification:**

The authors responded and provided details on image backbone architecture and replanning. These are important points which should be added to the main text. After reading the rebuttal and other reviewers comments I stand by "accept". Solid work with potentially high impact.

**Limitations:**

yes

**Paper Formatting Concerns:**

no concerns

**Quality:**

4

**Strengths And Weaknesses:**

Strengths:
- a solution to an important real-world robotic problem is proposed
- the solution is relatively simple and does not depend on heavy VLM compute at inference time
- experimental validation consists of tests both in simulation in real world and shows high success rate. I find fascinating how state-of-the-art VLM are able to achieve reasonable performance on an unseen robotics task, yet cannot outperform a task dedicated solution, reinforcing the "jack of all trades, master of none" claim for LLM.
- the system runs at 6-21 FPS which is additionally impressive.

---

> ### Author Rebuttal · Authors · 2025-07-31
>
> We thank the reviewer for the detailed review as well as the suggestions for improvement. Our response to the reviewer’s comments is below:
>
> **Q1: Details on the image backbone architecture.**
>
> We thank the reviewer for pointing out this omission. We will clarify the architecture of our image backbone in the revised version of the paper.
>
> To directly answer the question, we use the DLA-34 network [1] as our image backbone. We selected DLA-34 for its strong balance of computational efficiency and feature extraction capability, which is crucial for real-time robotics applications. The primary role of the backbone is to transform the input image into a rich geometric representation of the scene. Specifically, it is trained to produce a heatmap of object centers [2], regress object 3D dimensions, and predict their 3D orientations using a robust classification-then-regression method [3]. Our image backbone provides a rich, geometric understanding of the environment that is essential for the subsequent ordering task.
>
> We note that our framework is not a simple pipeline where detection is followed by a separate ordering step. Instead, the ordering module is tightly integrated with the perception backbone. It operates directly on the deep feature maps from DLA-34. By using an attention mechanism on these rich features, our ordering module can reason about the complex spatial and contextual relationships between objects. This integrated design is key to our method's ability to determine a feasible manipulation order and distinguishes it from approaches that rely only on post-processed detection results.
>
> While DLA-34 provides a strong and efficient baseline, we also note that the architectural principles of our method are general, and we anticipate that its performance could be further enhanced by leveraging more advanced backbones.
>
> ---
>
> **Q2: Replanning strategy and plan stability.**
>
> We are grateful to the reviewer for this insightful question about our system's planning strategy and robustness.
> To directly answer the first part of the question, in the experiments reported in our main paper, OrderMind replans after each manipulation action is completed. We chose this strategy to maximize robustness against unexpected perturbations common in real-world interactions. For instance, manipulating one object can slightly shift others and alter the scene's state. Since our method is real-time, replanning after each action allows the system to always base its next move on the most current environmental state without incurring significant computational overhead.
>
> To explore the stability of our generated plans, we conducted additional simulation studies to evaluate the system's performance with less frequent replanning. Specifically, we evaluated performance at replanning intervals of 1, 5, and 10 manipulations. We compared our unified model with the two-stage detection-then-sort method in terms of ID shuffle and task success rate. We adapted our Spatial Priors, which were described in Sec. 3.3 in our main paper, into a heuristic policy and name it as Spatial Priors Heuristic (SPH).
>
> To quantify the degree of index shuffling between consecutive manipulation plans, we employed the Levenshtein Distance (LD). LD is a classic metric that measures how dissimilar two sequences are by counting the minimum required edits (insertion, deletion, or substitution). For instance, since swapping two elements requires two distinct edits so that $LD([A, B, C], [A, C, B]) = 2$. This characteristic makes the metric highly suitable for assessing structural changes within a sequence. We present the results in the table below, comparing the success rate and Levenshtein Distance on the moderate difficulty scenes.
>
> **Table 1.** Levenshtein Distance (LD) and success rate (SR) under different replanning intervals. "Privileged" indicates methods with access to ground-truth object intrinsic and extrinsic attributes.
>
> | Replanning Interval |    Framework            | Privilege | LD ↓ | SR ↑ |
> | :------------------ | :------------------- | :-------: | :--: | :--: |
> | **1**         | GT + SPH             |     √     | 1.5  | 96.3 |
> |                     | YOLO11-seg + SPH     |     ×     | 3.6  | 77.5 |
> |                     | YOLO11-det + SPH     |     ×     | 1.8  | 79.8 |
> |                     | **OrderMind (Ours)** |     ×     | 1.7  | 89.6 |
> | **5**         | GT + SPH             |     √     | 5.2  | 85.5 |
> |                     | YOLO11-seg + SPH     |     ×     | 6.6  | 75.3 |
> |                     | YOLO11-det + SPH     |     ×     | 6.5  | 72.2 |
> |                     | **OrderMind (Ours)** |     ×     | 3.5  | 82.3 |
> | **10**        | GT + SPH             |     √     | 7.5  | 85.5 |
> |                     | YOLO11-seg + SPH     |     ×     | 9.3  | 70.6 |
> |                     | YOLO11-det + SPH     |     ×     | 9.3  | 69.0 |
> |                     | **OrderMind (Ours)** |     ×     | 4.7  | 80.5 |
>
> While the heuristic-based baselines without privileged information may result in higher plan instability and a drop in success rate, our method maintains more stable plans and a higher success rate. After several manipulation steps without replanning, heuristic methods exhibit larger ID reshuffling, as reflected by higher Levenshtein distances between consecutive plans. Across all tested replanning intervals, our approach consistently outperforms the two-stage detection-then-sort method, achieving both a higher success rate and lower Levenshtein distances.
>
> We attribute this to our unified design of jointly learning spatial representation and manipulation ordering. Heuristic methods are often greedy, i.e., they focus on the best immediate action based on local object stacking relationships, but they struggle to foresee the long-term consequences of an action. In contrast, our method distills the knowledge from large vision-language models and integrates rich visual and geometric context to generate a more globally coherent and far-sighted plan. This makes our plans inherently more robust and stable over longer execution horizons, leading to less index shuffling and a more gradual degradation in performance in few replan settings.
>
> **Reference**
>
> [1] Yu F, Wang D, Shelhamer E, et al. Deep layer aggregation. CVPR2018
>
> [2] Zhou X, Koltun V, Krähenbühl P. Tracking objects as points. ECCV2020
>
> [3] Guha E K, Natarajan S, Möllenhoff T, et al. Conformal Prediction via Regression-as-Classification. ICLR2024

---

### Note · Authors · 2025-08-12

We sincerely thank reviewers for their insightful feedback and engaging discussions.

We are delighted our rebuttal was well-received. Reviewers **Erw7** & **84U8** provided positive feedback. We also thank Reviewer **7BAs** for acknowledging our new results. To address final concerns on failure case and task analysis, we will integrate a deeper diagnosis in the revised version. For Reviewer **RBro**, we have provided clarifications on all major points and respectfully await the updated assessment.

Our primary contribution is **a unified, spatial-aware model for manipulation ordering**. In our architecture, the ordering module is tightly integrated with a perception image backbone and reasons directly on rich visual feature maps, allowing the model to holistically perceive and reason about the scene.

This design yields clear advantages:
1. **Superior Performance & Efficiency:** Our unified model significantly outperforms the two-stage detect-then-sort framework by **+14.0%** in hard scenarios and achieves about **2x faster** inference speed, demonstrating the practical benefits of its integrated architecture.
2. **Scalable Supervision via VLM**: We show that simple heuristics (e.g., height-based) fail in commonly-seen tilted supported scenarios. Instead of keeping engineering sophisticated rules, we pioneer the use of **VLM-generated ordering labels as a fundamentally more scalable and physically-grounded supervision source**, effectively distilling complex reasoning into our model.
3. **Long-Horizon Planning**: Our model demonstrates **long-horizon planning by generating stable manipulation plans**. Compared to heuristic baselines, our plans have lower Levenshtein distance and lead to higher success rates, especially in scenarios with infrequent replanning.
4. **Strong Generalizability**: Our model demonstrates robustness, **maintaining high performance across novel objects, unseen scenarios, and even moderate label noise**, providing evidence of its viability for real-world deployment.

In summary, our work presents a robust, efficient and well-designed solution for intelligent embodied robotics. We are confident it will be a valuable contribution to the community.

---

### Decision · Program_Chairs · 2025-09-17

**Decision:**

Accept (poster)

**Comment:**

This paper focuses on an interesting and specific manipulation task: object ordering in cluttered environments. This research holds practical significance for robotic manipulation.

Extensive experiments conducted in both simulated and real-world environments demonstrate that the proposed method significantly outperforms prior approaches in terms of both effectiveness and efficiency.

One reviewer votes a borderline reject, primarily raising concerns about the necessity of the proposed method when compared to a simple pipeline. While a simple pipeline is indeed feasible in straightforward scenarios, it suffers from a low performance ceiling and cannot be easily scaled to complex environments. Moreover, the proposed method does demonstrate superior performance to LM-based methods.

The AC recommends accepting this submission.